https://doi.org/10.1038/s41467-019-10327-5　　**OPEN**

# NREM sleep in the rodent neocortex and hippocampus reflects excitable dynamics

Daniel Levenstein [1,2], György Buzsáki [1,2] & John Rinzel[1,3]

During non-rapid eye movement (NREM) sleep, neuronal populations in the mammalian forebrain alternate between periods of spiking and inactivity. Termed the slow oscillation in the neocortex and sharp wave-ripples in the hippocampus, these alternations are often considered separately but are both crucial for NREM functions. By directly comparing experimental observations of naturally-sleeping rats with a mean field model of an adapting, recurrent neuronal population, we find that the neocortical alternations reflect a dynamical regime in which a stable active state is interrupted by transient inactive states (slow waves) while the hippocampal alternations reflect a stable inactive state interrupted by transient active states (sharp waves). We propose that during NREM sleep in the rodent, hippocampal and neocortical populations are excitable: each in a stable state from which internal fluctuations or external perturbation can evoke the stereotyped population events that mediate NREM functions.

[1] Center for Neural Science, New York University, 4 Washington Pl, New York, NY 10003, USA. [2] NYU Neuroscience Institute, 450 East 29th Street, New York, NY 10016, USA. [3] Courant Institute for Mathematical Sciences, New York University, 251 Mercer St, New York 10012, USA. Correspondence and requests for materials should be addressed to J.R. (email: rinzeljm@gmail.com)

Sleep function relies on internally-generated dynamics in neuronal populations. In the neocortex, non-rapid eye movement (NREM) sleep is dominated by a "slow oscillation"[1]: alternations between periods of spiking (UP states) and periods of hyperpolarization (DOWN states) that correspond to large "slow waves" in the local field potential (LFP)[2,3] (Fig. 1a, b, Supplementary Fig. 1). In the hippocampus, NREM sleep is dominated by sharp wave-ripple dynamics: periods of spiking (SWRs) separated by periods of relative inactivity (inter-SWRs)[4] (Fig. 1e, f). The functional importance of these dynamics is well established: slow waves and SWRs perform homeostatic maintenance of the local synaptic network in both regions[5–7], and their temporal coupling[8–11] supports the consolidation of recently-learned memories[12–14]. However, it's unclear how the state of neuronal populations in the two regions promotes the generation of their respective dynamics, or how population state supports the propagation of neural activity between structures.

To study the state of hippocampal and neocortical populations during NREM sleep, we used an idealized model of an adapting recurrent neuronal population (Fig. 1c–g). Similar models have been directly matched to neocortical UP/DOWN alternations during anesthesia and in slice preparations[15–17]. These studies found that the alternations in slice are adaptation-mediated oscillations[16], while those under anesthesia reflect noise-induced switches between bistable states[15]. However, neuronal dynamics during NREM are distinct from those in anesthesia/slice[18]. We show how a few physiological parameters can determine the properties of alternation dynamics in neuronal populations, and

identify parameter domains that match experimental data from naturally-sleeping rats[19]. This treatment revealed that neocortical and hippocampal alternation dynamics can be explained using the same model, in complementary regimes of activity.

We report that during NREM sleep the rodent neocortex and hippocampus are neither endogenously oscillatory nor bistable, but are excitable: neural populations in each region rest in a stable state from which suprathreshold fluctuations can induce transient events that are terminated by the influence of adaptation. Specifically, the neocortex maintains a stable UP state with fluctuation-induced transitions to a transient DOWN state (slow waves), while the hippocampus rests in a stable DOWN state with fluctuation-induced transitions to a transient UP state (SWRs). Each region can generate its respective population event spontaneously (due to internally-generated fluctuations) or in response to an external perturbation (such as input from another brain structure). As a result, alternations in both structures show asymmetric duration distributions (Fig. 1d–h). We further observe that variation in the depth of NREM sleep corresponds to variation in the stability of the neocortical UP state. Our findings reveal a unifying picture of the state of hippocampal and neocortical populations during NREM sleep, which suggests that NREM function relies on excitable dynamics in the two regions.

## Results

### UP/DOWN dynamics in an adapting excitatory population model. UP/DOWN alternations are readily produced in models

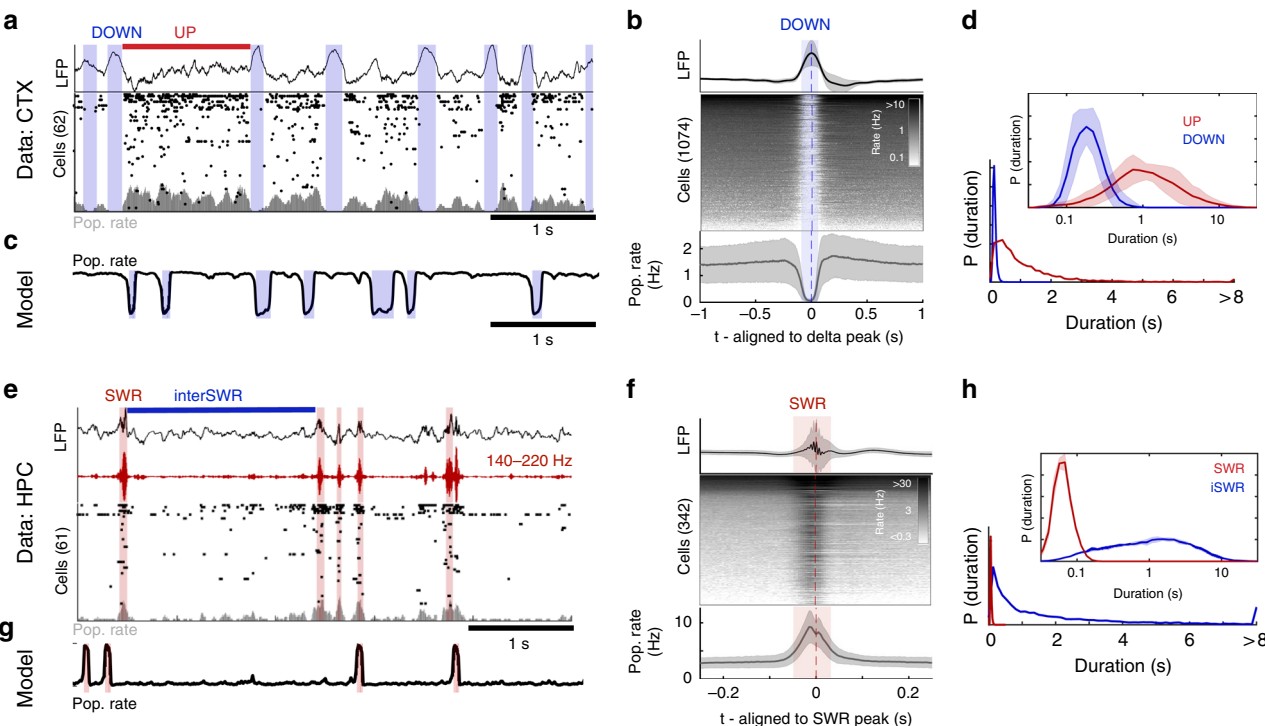

**Fig. 1** Neocortical UP/DOWN and hippocampal SWR dynamics during NREM sleep. **a** A sample of data from rat mPFC during NREM sleep. Data were collected using high-density silicon probes[19]. LFP and spike times from cortical neurons were extracted as reported previously (see Methods). Neocortical slow waves are coincident with population-wide non-spiking DOWN states, which alternate with UP states of longer duration. **b** Peri-event time histogram aligned to delta peaks for the LFP (top), all recorded cells (middle), and population rate (bottom). **c** Simulation of the model (Eqns 1–2). Parameters determined by matching in vivo and simulated UP/DOWN state dwell times, as described in section: 'Neocortex is in an Excitable_UP regime during NREM sleep'. **d** UP/DOWN state dwell time distribution (bottom) in linear (example recording) and logarithmic scale. **e** A sample of data from rat CA1 (HPC) during NREM sleep[65]. Detected sharp wave-ripples (SWR) indicated with red. **f** Peri-event time histogram aligned to SWR peaks for the LFP (top), all recorded cells (middle), and population rate (bottom). **g** Simulated r-a model with the best-matching parameters, as described in section: 'Hippocampus is in an Excitable_DOWN regime during NREM sleep'. **h** SWR and inter-SWR duration distributions in linear (example recording) and logarithmic scale. All shaded lines reflect mean ± standard deviation over recordings

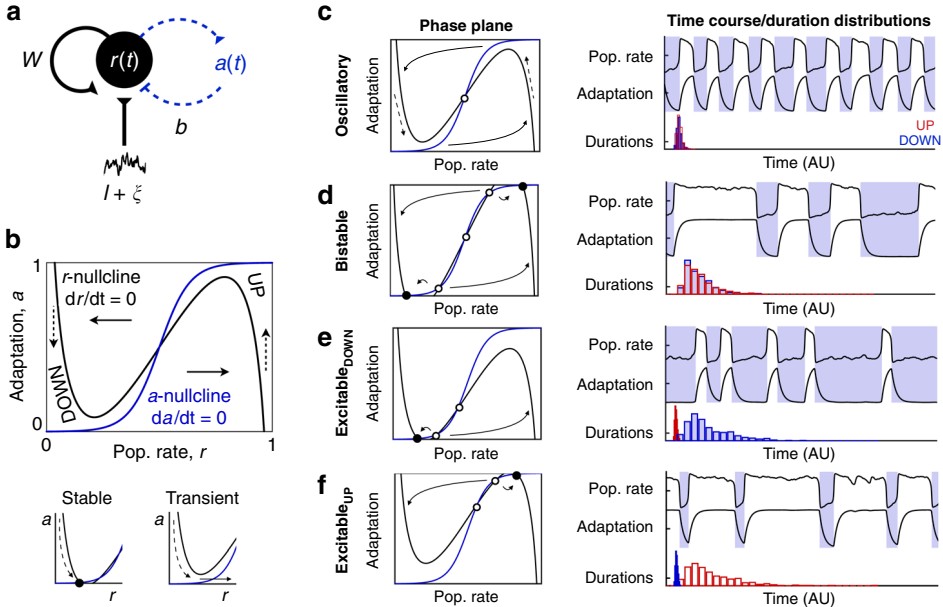

**Fig. 2** UP/DOWN dynamics in an adapting recurrent neural population. **a** Wilson–Cowan-like model for a neural population with slow adaptive process. **b** $r$-$a$ Phase plane. Model dynamics are seen as trajectories in the phase plane that follow Equations (1–2). Dashed arrows indicate slow vertical trajectories at timescale of $\tau_a$, solid arrows indicate fast horizontal trajectories at timescale of $\tau_r$. Nullclines (the two curves along which $dr/dt = 0$, $da/dt = 0$) and their intersections graphically represent dynamics for a given set of parameter values (parameters as defined in C shown). Left and right branches of the $r$-nullcline correspond to DOWN and UP states, respectively. Stable UP/DOWN states are seen as stable fixed points at nullcline intersections. Transient UP/DOWN states are seen as $r$-nullcline branch with no intersection. **c**–**f** Four UP/DOWN regimes available to the model, as distinguished by location of stable fixed points (see also Supplementary Fig. 3). Representative phase plane (Left), simulated time course and UP/DOWN state duration distributions (right, time units arbitrary) for each regime. Stable fixed points are represented by filled circles, unstable fixed points by empty circles. Parameters: (**c**–**f**) $b = 1$, (**c**, **e**, **f**) $w = 6$ (**d**) $w = 6.3$, (**c**) $I = 2.5$ (**d**) $I = 2.35$ (**e**) $I = 2.4$ (**f**) $I = 2.5$. Default parameters specified in methods

of neural populations with recurrent excitation and slow adaptive feedback[15–17,20–25]. We first consider the dynamics of a model in which neuronal population activity is described in terms of the mean firing rate, $r(t)$, subject to adaptation, $a(t)$ (Fig. 2a).

$$\tau_r \frac{dr}{dt} = -r + R_\infty(wr - ba + I + \xi(t)) \qquad (1)$$

$$\tau_a \frac{da}{dt} = -a + A_\infty(r) \qquad (2)$$

Equations (1–2) describe how $r$ and $a$ evolve in time as a function of the net input to the population: the sum of the recurrent excitation with weight $w$ and a background level of drive with a tonic parameter $I$, and noisy fluctuations $\xi(t)$, minus adaptation weighted by gain parameter $b$. $R_\infty(\text{input})$ is a function that defines the population rate given constant net input. Similarly, $A_\infty(r)$ defines the level of adaptation given a fixed rate. To enable the analytical treatment of model dynamics in the following section, both $R_\infty(\text{input})$ and $A_\infty(r)$ are taken to be sigmoidal functions. However, this choice is not critical for the generality of our findings. Further model details and physiological interpretation of parameters can be found in Supplementary Note 1 and the Supplementary Discussion.

Model dynamics can be represented as a trajectory in the $r$–$a$ phase plane[26] (Fig. 2b, Supplementary Note 1). Steady states, or fixed points, of activity are found at intersections of two curves defined by the conditions $\frac{dr}{dt} = 0$ and $\frac{da}{dt} = 0$, the $r$- and $a$-nullclines. Depending on parameter values, the model can show four distinct regimes of UP/DOWN dynamics—distinguished by whether UP/DOWN transitions are noise- or adaptation-induced, and thus the stability or transient nature of the UP and DOWN states (Fig. 2b)[15,16].

In the oscillatory regime (Fig. 2c), activity alternates between transient UP and DOWN states at a relatively stable frequency. Adaptation activates during the UP state and brings the population to the DOWN state, during which adaptation inactivates and the population returns to the UP state. Because $r(t)$ is fast compared to the slow adaptation, the $r(t)$ time course and the phase plane trajectory are square-shaped, with rapid transitions between UP and DOWN states.

If the UP and DOWN state are both stable, the system is bistable (Fig. 2d). In this regime, adaptation is not strong enough to induce UP/DOWN state transitions. However, sufficiently large (suprathreshold) fluctuations can perturb the population activity to cross the middle branch of the $r$-nullcline, resulting in a transition to the opposing branch. Thus, the presence of noise induces alternations between UP and DOWN states, resulting in highly variable UP/DOWN state durations.

In the case of a single stable state, the system can show UP/DOWN alternations in one of two excitable regimes. If the DOWN state is stable (Fig. 2e), the system is in an Excitable_DOWN regime. The population will remain in the DOWN state in the absence of any external influence. However, a brief suprathreshold activating input can trigger a rapid transition to a transient UP state, during which adaptation activates, leading to a return to the DOWN branch. In the presence of noise, UP states are spontaneously triggered by net activating fluctuations. The time course of the model in the Excitable_DOWN regime shows long DOWN states of variable durations punctuated by brief stereotyped UP states.

Conversely, if the UP state is stable, the system is in an Excitable_UP regime (Fig. 2f). Brief inactivating input can elicit a switch from the UP state to a transient DOWN state, during which adaptation deactivates, leading to a return to the UP branch. In the presence of noise, DOWN states are spontaneously

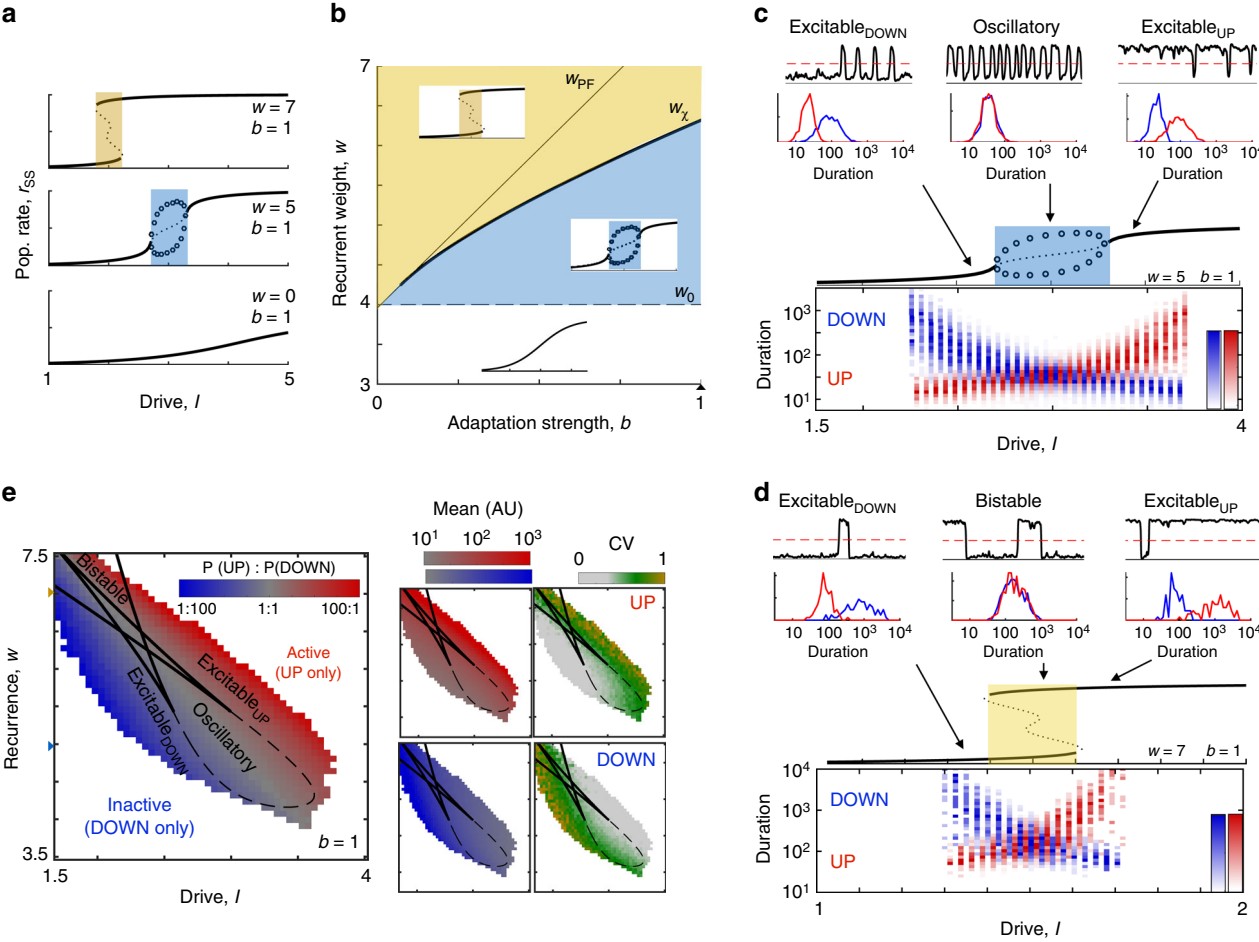

**Fig. 3** Recurrence, adaptation, and drive determine dynamical regime and duration statistics. **a** Effective I/O curves for populations with increasing levels of recurrent excitation, $w$. Solid lines indicate stable fixed points, dotted lines indicate unstable fixed points, and circles indicate the upper/lower bounds of a limit cycle. Yellow indicates bistable regime (2 stable fixed points), blue indicates oscillatory regime (0 stable fixed points with a stable limit cycle). See also Supplementary Figs. 3 and 4. **b** Regions in the $w$-$b$ parameter plane for distinct I/O properties, indicated by insets. Blue shading indicates parameter values for a oscillatory-centered I/O curve, yellow shading indicates a bistable-centered I/O curve. Boundary curves correspond to parameters where bifurcations vs $I$ coalesce, see Supplementary Note 3. $w_{PF}$: pitchfork bifurcation separating 5-fixed point and 3-fixed point (2 stable) regimes. $w_\chi$: degenerate pair of saddle node bifurcations, separating bistable-centered and oscillatory-centered I/O curves. $w_0$: degenerate Hopf bifurcation separating oscillatory-centered and monotonic stable I/O curves. **c** Simulated duration distributions on the oscillatory-centered I/O curve. (Top) Simulated time course ($r$ vs $t$) and dwell time distributions (bottom) for low ($I = 2.6$), intermediate ($I = 3$), and high ($I = 3.4$) levels of drive. (Bottom) UP/DOWN state duration distributions as a function of drive to the system. Note logarithmic axis of duration. Color reflects observed proportion of UP/DOWN states with a given duration. **d** Same as **c**, for the bistable-centered I/O curve. **e** Duration statistics as a signature of UP/DOWN regime. (Left) Ratio of simulation time in UP and DOWN state. (Right) Mean and CV of simulated dwell times for UP and DOWN states in the I-W parameter space with fixed $b$ (triangle in 3B)

triggered by net-inactivating fluctuations. The time course will show longer UP states of variable durations with stereotypically brief DOWN states. These two regimes (Fig. 2e, f) are excitable because relatively small fluctuations in population rate can excite the population out of a stable steady state and induce disproportionately large, stereotyped, population events: a transient UP state in the case of the Excitable$_{DOWN}$ regime and a transient DOWN state in the case of the Excitable$_{UP}$ regime.

**Recurrence, adaptation, and drive control UP/DOWN regimes.** How do the properties of a neuronal population determine dynamical regime? We use numerical and analytical methods from dynamical systems theory[26] to reveal how intrinsic and network properties determine the properties of UP/DOWN dynamics in our model. The analysis is summarized here and presented in further detail in Supplementary Notes 2–3 and Supplementary Figs. 2–4.

We first consider the population's effective input/output relation (I/O curve): how the population rate fixed points, $r_{ss}$, depend on the level of drive (Fig. 3a). If recurrence is weak, the I/O curve increases monotonically with drive and no UP/DOWN alternations are possible. At a critical value of recurrent excitation the population is able to self-maintain an UP state (Supplementary Note 2, Supplementary Fig. 2), and UP/DOWN alternations emerge between low-rate activity at weak drive and high-rate activity at strong drive. Recurrence and adaptation oppositely influence the dynamical regime at the I/O curve's center region (Supplementary Fig. 3). By identifying parameter values at which transitions occur in the dynamical regime at the half-activation point of the I/O curve (Fig. 3b, Supplementary Note 3, Supplementary Fig. 4), we see that the population will have an oscillatory-centered I/O curve with stronger adaptation (Fig. 3b, blue) and a bistable-centered I/O curve with stronger recurrence (Fig. 3b, yellow).

In the absence of noise or external perturbation, only the oscillatory regime will alternate between UP and DOWN states. We next consider the effects of noise on an oscillatory-centered I/O curve (Fig. 3c). Within the oscillatory regime, the simulated population rate alternates regularly between transient UP and DOWN states, and UP/DOWN state durations reflect the time scale of adaptation, $\sim\tau_a$ (Supplementary Fig. 5). For $I$-values above the oscillatory regime, noise can evoke transitions from the stable UP state to a transient DOWN state (an Excitable$_{UP}$ regime). DOWN state durations still reflect the time scale of adaptation, $\tau_a$, but UP state durations now reflect the waiting time for random fluctuations to drop the system out of the UP state attractor, and thus vary with noise amplitude (Supplementary Fig. 5). As drive is further increased, the effective stability of the UP state increases and larger fluctuations are needed to end the UP state. Thus, UP states become progressively longer, while DOWN states stay approximately the same duration ($\sim\tau_a$). The same case is seen for values of $I$ below the oscillatory regime but with UP/DOWN roles reversed (i.e., an Excitable$_{DOWN}$ regime). Similar response properties are seen for the bistable-centered I/O curve (Fig. 3d). In both cases, the duration distributions plotted vs. drive form a crossed-pair, with a center symmetrical portion (i.e., an oscillatory (Fig. 3c) or bistable (Fig. 3d) regime) flanked by the asymmetrical Excitable$_{DOWN}$ and Excitable$_{UP}$ regimes.

We next expand our analysis of simulated duration distributions to a representative $I$–$w$ parameter plane ($b = 1$, Fig. 3e) and $I$–$b$ plane ($w = 6$, Supplementary Fig. 5). The mean durations vary continuously as the level of drive brings the population from a DOWN-dominated to an UP-dominated regime. However, the duration variability (as measured by the coefficient of variation, CV) shows sharp transitions at the boundaries between regimes, which reflect the different mechanism of transitions out of stable and transient states. In general, the durations of stable states are longer and more variable, while those of transient states are shorter and less variable. Thus, the statistics of UP/DOWN state durations reflect the underlying dynamical regime, allowing us to effectively distinguish oscillatory, bistable, and excitable dynamics.

**Neocortex is in an Excitable$_{UP}$ regime during NREM sleep**. The durations of neocortical UP/DOWN states (Fig. 1) are indicative of an Excitable$_{UP}$ regime in our model. Neocortical UP states during NREM are longer (mean$_{UP}$: $1.7 \pm 0.92$ s) compared to DOWN states (mean$_{DOWN}$: $0.21 \pm 0.05$ s), and more irregular (CV$_{UP} = 1.1 \pm 0.27$; CV$_{DOWN} = 0.38 \pm 0.06$) (Fig. 4a, all values mean $\pm$ std over recordings) suggesting a stable UP and transient DOWN state. We directly compared the simulated and experimentally-observed dynamics by matching the statistics of experimental UP/DOWN durations to those in Fig. 3e and Supplementary Fig. 5. We found that the region of parameter space in which the CV$_{UP}$, CV$_{DOWN}$ and ratio of mean durations is within 2 standard deviations of the experimental durations is in the Excitable$_{UP}$ regime (Fig. 4b, Supplementary Fig. 7 red outline). We next compared the shapes of the duration distributions between model and experiment. For each model realization (i.e., each point in the $I$–$w$ parameter plane), we calculated the similarity between simulated and experimental duration distributions for each recording session in the experimental dataset (Supplementary Figs. 6, 7, Methods). The domain of high similarity between animal data and the model fell in the Excitable$_{UP}$ regime, as indicated by the 25 best fit points and in the average value of similarity (over all 25 sessions) in $I$–$w$ parameter space (Fig. 4b) and in the $I$–$b$ parameter space (Supplementary Fig. 7). The simulated time course (Fig. 4d) and duration distributions (Fig. 4c) using the parameter set with highest mean similarity

over all sessions revealed a good match between experimental and modeled dynamics. The domain of high similarity was degenerate and remained in the Excitable$_{UP}$ regime with variation in the fixed parameters, $\tau_a$, $b$, and the amplitude of the noise (Supplementary Fig. 7). We thus found that NREM sleep in the rodent neocortex is characterized by an Excitable$_{UP}$ regime: a stable UP state with noise-induced transitions to a transient DOWN state.

**Hippocampus is in an Excitable$_{DOWN}$ regime during NREM sleep**. Since the burst-like dynamics of SWR is reminiscent of the Excitable$_{DOWN}$ regime of our model, we asked whether these patterns could also be explained by the same principles. InterSWR durations are much longer (mean = $2.0 \pm 0.22$ s) compared to SWR events (mean = $0.06 \pm 0.005$ s), and more variable (CV$_{InterSWR} = 1.3 \pm 0.10$; CV$_{SWR} = 0.33 \pm 0.04$) (Fig. 4e) suggesting a stable DOWN and transient UP state (SWR). We applied the duration distribution matching procedure to the SWR/inter-SWR duration distributions and confirmed that the $r$–$a$ model can also mimic SWR dynamics, with a band of high data-model similarity in the Excitable$_{DOWN}$ regime (Fig. 4g). Interestingly, our idealized model is not able to capture the short-interval inter-SWR periods associated with occasional SWR bursts (Supplementary Fig. 7), which suggest the presence of separate SWR-burst promoting mechanisms, possibly arising from interactions with the entorhinal cortex or spatially traveling patterns of SWRs in the hippocampus[27,28]. Accordingly, while the mean ratio and CV$_{SWR}$ of the best fitting model regime were within 2.5 standard deviations of those observed in vivo, the CV of inter-SWR periods was larger than expected from the model (i.e., CV > 1). This finding suggests that during NREM sleep the hippocampus is in a stable DOWN-like state, from which internal 'noise' or an external perturbation can induce population-wide spiking events.

**NREM depth corresponds to UP state stability**. For our initial analysis of the neocortical NREM data, we assumed that model parameters were stationary over the course of a sleep session. However, rodent NREM sleep has been classified on a spectrum from light to deep NREM, with higher power in the LFP delta band (1–4 Hz) reflecting deeper NREM sleep[29]. To investigate the relationship between changes in cortical state with NREM depth and UP/DOWN dynamics, we calculated the level of delta power in the 8 s time window surrounding each UP and DOWN state (Fig. 5a). UP state durations varied systematically with delta power (Fig. 5a–c, Supplementary Fig. 8): epochs of lower delta power contained longer UP states, and epochs of higher delta power were associated with shorter UP states (Fig. 5a–c, Supplementary Fig. 8). However, DOWN state durations were invariant with delta power, and the CV of UP state durations was consistently higher than DOWN state durations, as would be expected for Excitable$_{UP}$ dynamics with noise-induced transitions from a stable UP to a transient DOWN state.

We then grouped the experimental UP/DOWN states by delta power and calculated data-model similarity maps for UP/DOWN state durations in each group (Fig. 5d, Supplementary Fig. 8). We found that the vast majority of time in all recording sessions was spent in the parameter domain of the Excitable$_{UP}$ regime (Fig. 5b, bottom). However, with higher delta power, the best fitting model parameters moved closer to the transition to the oscillatory regime, and the epochs of highest delta power were well-matched by oscillatory dynamics in a small number of sessions.

**Evoked slow waves from an inhibition-stabilized UP state**. Our previous analyses considered a constant (stationary) source of noise that produced spontaneous transitions out of stable states in

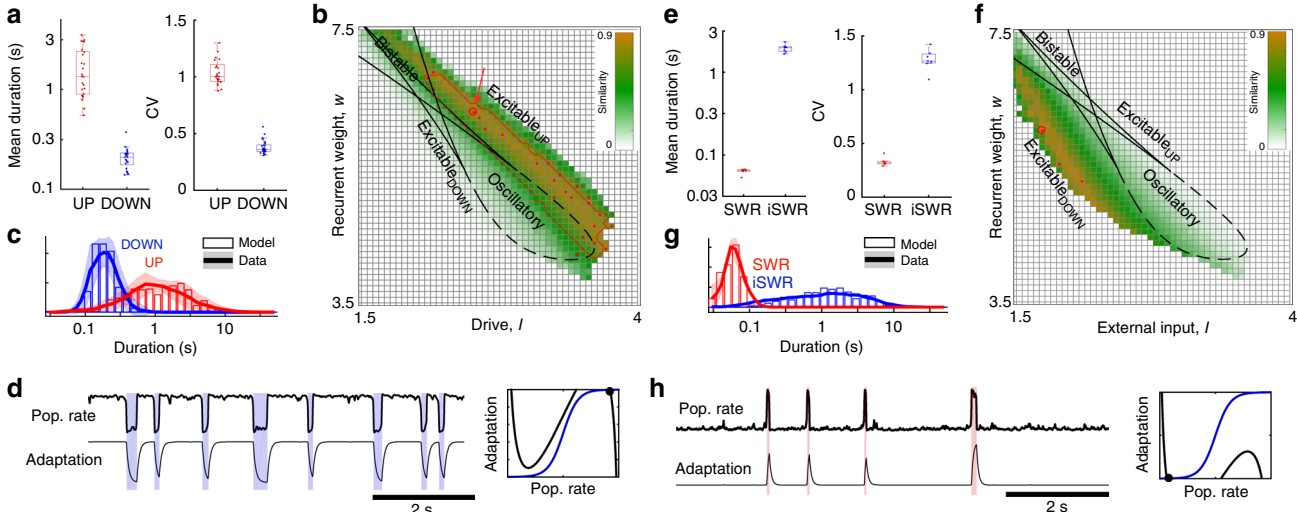

**Fig. 4** Neocortex and hippocampus are in excitable regimes during NREM sleep. **a** Mean and CV of UP/DOWN state durations, each point is a single recording session. **b** Similarity between simulated and experimental duration distributions (see Supplementary Fig. 6, Methods for similarity calculation). Color indicates mean similarity over all recordings, red line outlines region for which model simulations fall within mean ± 2STD of experimentally observed $CV_{UP}$, $CV_{DOWN}$ and $mean_{UP}/mean_{DOWN}$ ratio. 25 red dots indicate the most similar parameters for each of the 25 recordings, and large circle (arrow) is best mean similarity. **c** Neocortical duration distributions for data and model simulation in the best mean similarity parameters indicated in **b**. ($I = 2.64$, $W = 6.28$, $b = 1$). **d** Simulated r-a model with the best-matching parameters. (Right) Phase plane diagram for model with best-fit parameters, in the Excitable$_{UP}$ regime. **e** Mean and CV of SWR/interSWR state durations, each point is a single recording session ($n = 7$ hippocampal sessions). **f** Similarity between simulated and experimental duration distributions as in **b**. **g** Hippocampal duration distributions for data and model simulation in the best mean similarity parameters. **h** Simulated r-a model with the best-matching parameters indicated in **f**. ($I = 1.9$, $W = 6$, $b = 1$). (Right) Phase plane diagram for model with best-fit parameters, in the Excitable$_{DOWN}$ regime. Box plots center line: median, box limits: upper/lower quartiles, whiskers: data range (excluding outliers). Shaded lines reflect mean ± standard deviation over recordings

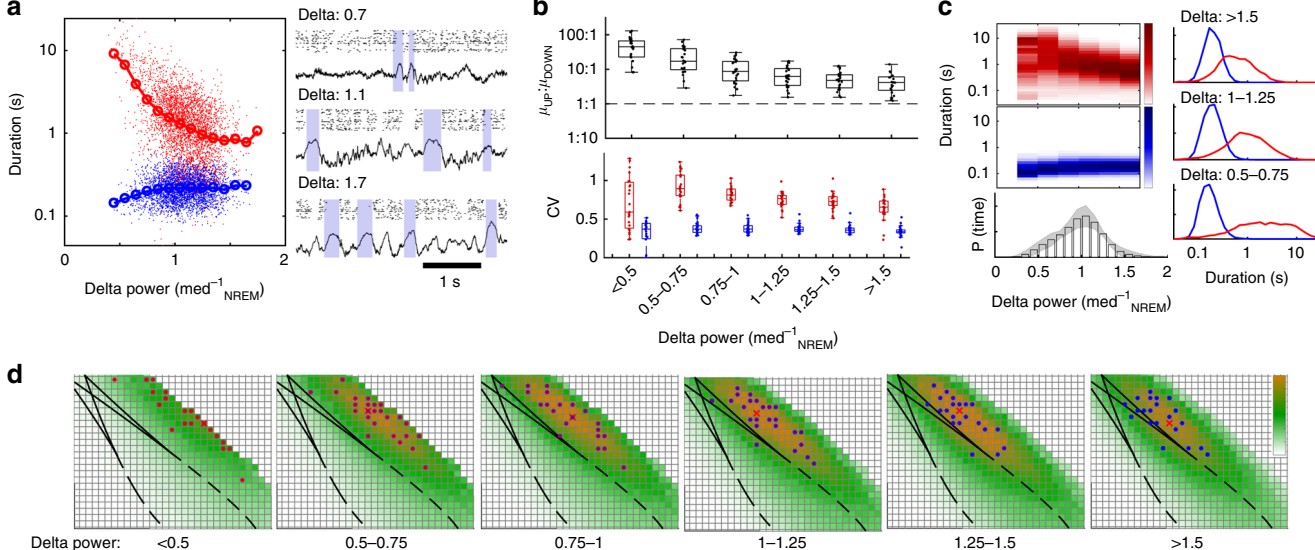

**Fig. 5** State-dependent variation in alternation dynamics during NREM sleep. **a** UP/DOWN state durations as a function of delta (1–4 Hz) power in a representative recording (delta power normalized to median power during NREM sleep). **b** (Top)Ratio of mean UP/DOWN durations in each neocortical recording as a function of delta power. (Bottom) CV of UP/DOWN durations in each neocortical recording as a function of delta power. Center line: media, box limits: upper/lower quartiles, whiskers: data range (excluding outliers). **c** (Top) Mean distribution of UP and DOWN state durations as a function of delta power in the surrounding 8 s window. Color indicates the observed proportion of UP/DOWN states with a given duration. (Bottom) Mean delta power distribution over all recordings, (error shade: standard deviation). (Right) Mean UP/DOWN state duration distributions for high, medium, and low delta power. **d** Map of in vivo-model similarity (as in Fig. 4b, with $\tau_r$ fixed at $\tau_r = 5$ ms), for the delta-power groups as in **c**

our model. We now consider a brief input that evokes a transient event. For the hippocampal-like Excitable$_{DOWN}$ regime, a brief increase in drive will evoke a transient UP state, (i.e., a SWR, Supplementary Fig. 9). In the absence of noise, perturbations must be of sufficient magnitude (i.e., suprathreshold). With noise,

the probability to evoke a SWR increases with magnitude of the perturbation (Supplementary Fig. 9). A converse situation is apparent for the neocortical-like Excitable$_{UP}$ regime—a brief decrease in drive is able to evoke a transient DOWN state (i.e., a slow wave, Supplementary Fig. 9). However, as long-range

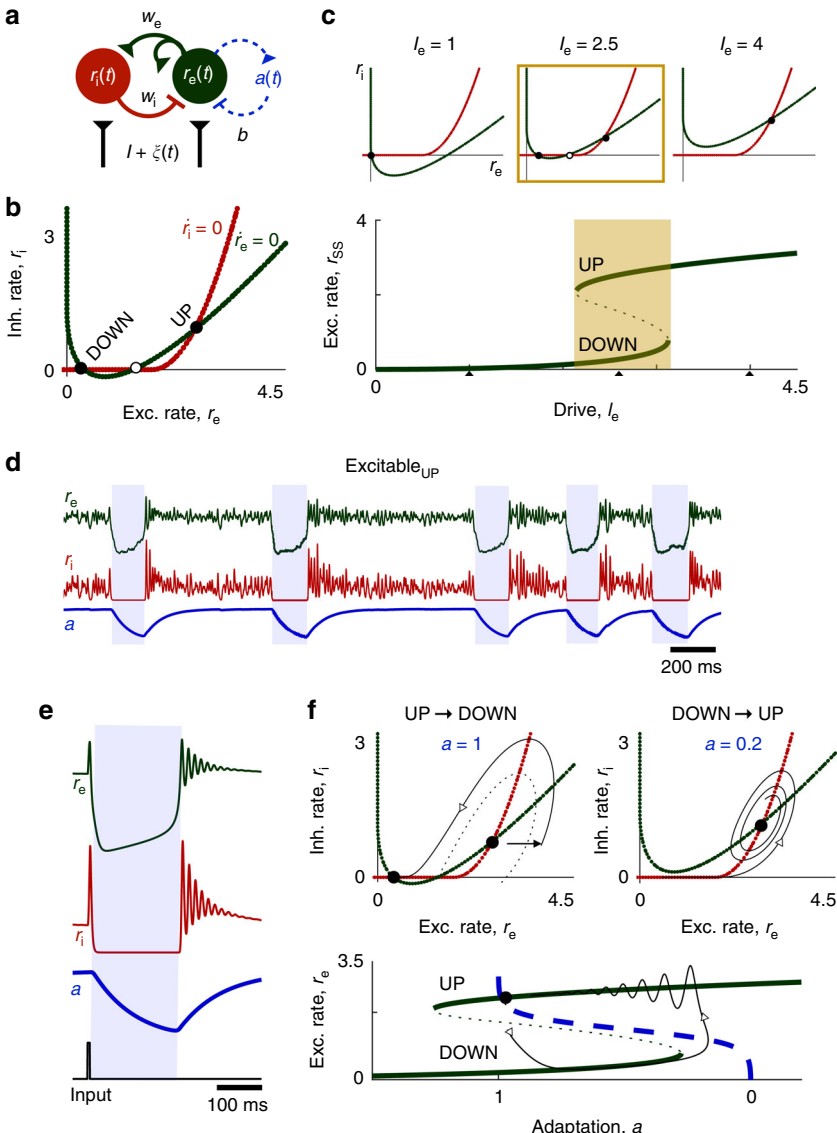

**Fig. 6** UP/DOWN dynamics in an Adapting Inhibition-Stabilized Network (aISN). **a** Model schematic. **b** $r_e$–$r_i$ phase plane with $a$ frozen. **c** Bistability in the aISN with $a$ frozen. Effective I/O curve of the excitatory population rate, $r_{ss}$, with $r_e$–$r_i$ phase planes at low, intermediate, and high levels of drive to the E population (Top). Parameters: $w_e = 4$, $w_i = 3$. **d** Excitable$_{UP}$ dynamics in the aISN. Parameters: $\tau_e = \tau_i = 5$ ms, $\tau_a = 200$ ms. **e** UP/DOWN transitions in the Excitable$_{UP}$ aISN. (Left) Time course of the model in response to brief excitatory input to the excitatory population. **f** (Bottom) Trajectory of the time course in the $r_e$–$a$ phase plane after stimulus-induced transition. The effective I/O curve, $r_{ss}$, and the steady state adaptation curve, $A_\infty(r)$, act like the $r$ and $a$ nullclines in the excitation only model. (Top) Trajectories of the stimulus-induced UP→DOWN and subsequent DOWN→UP transition in the $r_e$–$r_i$ phase plane. The dotted line separates the basin of attraction for the DOWN and UP state fixed points

projections tend to be excitatory, we wondered how an excitatory perturbation might evoke a neocortical UP→DOWN transition.

Neuronal spike rates during the UP state are generally low[19] with balanced excitatory and inhibitory synaptic inputs[30]. Previous work has shown that models with fast inhibition and slow adaptation can give UP/DOWN alternations in the same four regimes described above[15] with a low-rate UP state that is stabilized by feedback inhibition[31,32]. We hypothesized that local inhibitory cells may support excitation-induced UP→DOWN transitions, and included an inhibitory population ($\tau_i \approx \tau_e$) in the model (Fig. 6a):

$$\tau_e \frac{dr_e}{dt} = -r_e + R_{e,\infty}(w_{ee}r_e - w_{ei}r_i - ba + I_e + \xi_e(t)) \quad (3)$$

$$\tau_i \frac{dr_i}{dt} = -r_i + R_{i,\infty}(w_{ie}r_e - w_{ii}r_i + I_i + \xi_i(t)) \quad (4)$$

$$\tau_a \frac{da}{dt} = -a + A_\infty(r_e) \quad (5)$$

where adaptation acts on the excitatory population and $R_{e,\infty}(\text{input})$ and $R_{i,\infty}(\text{input})$ are threshold power law I/O relations, as seen in the in vivo-like fluctuation-driven regime[33] (Supplementary Note 4).

Given that adaptation is slow we can treat $a$ as frozen and visualize model dynamics in the $r_e$–$r_i$ phase plane (Fig. 6b). The fixed point value of $r_e$ as a function of drive describes the effective I/O curve of the network ($r_{ss}$, Fig. 6c). Like the excitation-only model, strong recurrent excitation induces bistability at low levels of drive (Supplementary Fig. 10). In the bistable condition, the $r_e$–$r_i$ phase plane shows stable UP and DOWN state fixed points, separated by a saddle point (Fig. 6b, c). With $a$ dynamic, the

model can have steady state fixed points on either the UP or the DOWN branch of the I/O curve, resulting in the same regimes as the two-variable model described above[15] (Supplementary Fig. 10).

We investigated Excitable$_{UP}$ dynamics in the adapting, inhibition-stabilized model[33] (Fig. 6d–f). Consider a transition from the UP to the DOWN state (Fig. 6f). As adaptation slowly deactivates, the system drifts along the DOWN branch. Eventually, the DOWN state loses stability, the trajectory reaches and rounds the lower knee of the I/O curve and transitions abruptly to the only remaining stable solution: the UP state. Adaptation then builds as the system returns to the stable UP state fixed point.

Due to the effects of inhibition, small perturbations from the UP state fixed point exhibit damped oscillations as the system returns to steady state. The damped oscillations arise from transient imbalance of excitation and inhibition, and occur when the UP state fixed point is an attracting spiral. As a result, high-frequency oscillations (at a time scale set by the excitatory and inhibitory time constants) occur at the DOWN→UP transition. A further implication is that excitatory input to the excitatory population (Fig. 6e) can recruit sufficient inhibition to force the entire network into a DOWN state. This threshold effect is seen as a trajectory in the phase plane that separates the basins of attraction of the UP and DOWN state (i.e., a separatrix, Fig. 6f). The separatrix emerges (in reverse time) from the saddle and curves around the UP state fixed point. From this visualization we see that a brief increase in the rate of either population can push the trajectory out of the UP state basin of attraction (Fig. 6f). Thus, a transient DOWN state (i.e. a slow wave) can be evoked by an excitatory perturbation, as well as drops in the excitatory population rate.

## Discussion

To account for cortical dynamics during NREM sleep, we used a firing rate model that represents a neuronal population with positive feedback (recurrent excitation) and slow negative feedback (adaptation). Although the model is idealized, it is amenable to mathematical treatment in terms of a few key parameters that allowed us to develop intuitions for the repertoire of dynamics available to an adapting, recurrent neural population. Our analysis revealed how the level of drive and the relative strength of recurrent excitation and adaptation create a spectrum of dynamical regimes with UP/DOWN alternations, defined by the stability or transience of UP and DOWN states (Fig. 7a). We found that both neocortical and hippocampal alternations during NREM sleep are well-matched by the model in excitable regimes that produce characteristically asymmetric distributions of UP and DOWN state durations. We next discuss implications of the findings for NREM sleep. Additional discussion on UP/DOWN alternations in other physiological contexts can be found in the Supplementary Discussion.

Despite the widely used term slow "oscillation"[1], the asymmetric duration distributions during NREM indicate that the NREM slow oscillation is aperiodic: reflecting a stable UP state from which ongoing activity fluctuations induce transient DOWN states (i.e., slow waves). A key feature of the model is the noise responsible for initiating spontaneous UP→DOWN transitions. In neuronal network modeling, noise often refers to unidentified fluctuations in physiological activity and can be divided into fluctuations internal to the population and fluctuations from afferent projections. While we do not explicitly distinguish them in the model, we assume that both sources play a role in initiating cortical UP→DOWN transitions. Population rate fluctuates during the UP state due to finite size effects[34] and

temporal correlations that emerge with strong recurrent connections[35]. Similarly, the level of afferent activity from thalmo- or cortico-cortical projections would be expected to fluctuate. We also note that while the isolated neocortex can produce UP/DOWN state alternations[36], we should consider the thalamo-cortical system for understanding slow wave dynamics in vivo[37]. Because the cortex and corresponding thalamic nuclei are highly interconnected, cortex and thalamus may transition UP and DOWN together and reflect interacting (as opposed to independent) systems. However, it was recently found that cortex tends to lead the thalamus into the DOWN state[38]. Future work should expand the model to include a thalamic population, which would also help to understand the interaction of slow waves with thalamocortical spindle oscillations[11,39,40].

While rodent sleep does differ from that seen in humans, both humans and rodents have slow waves with similar underlying physiology, and both species have NREM sleep of varying depth[29,41]. We found that the depth of NREM sleep in the rodent reflects the stability of the UP state in a manner that resembles the stages of NREM/SWS sleep in humans[41]. In light NREM sleep (human stage N1), long UP states are occasionally punctuated by neuronal silence-associated delta or slow waves, which can be localized at one or few recording sites across the cortical mantle[11]. As sleep deepens, the incidence of DOWN states increase and they become synchronous over larger cortical areas[42] (N2 stage). The DOWN-UP transitions occasionally become strongly synchronous, producing a sharp LFP wave known as the K complex[43]. With further deepening of sleep, DOWN states become more frequent and short episodes of repeating DOWN states may become rhythmic (N3/SWS stage). While the comparison between rodent and human sleep data was not performed, we found a similar evolution in rodent NREM. We found that the N3-like oscillatory state in the rat occupies only a small fraction of NREM sleep, whereas in humans this stage is more prominent. Our analysis predicts that deeper stages of NREM reflect a less stable UP state, which may be due to (1) decreased recurrent strength, (2) decreased neuronal excitability or (3) increased strength of adaptation[22,44]. We hope that this work can provide a framework to guide comparative studies on the differences between rodent and human sleep.

While the model is ambiguous to the biophysical substrate of adaptation, we can make some predictions: first, the adaptive process responsible for neocortical UP/DOWN alternations should be constitutively active during the UP state and deactivate during the hyperpolarized DOWN state. Second, the adaptive process should recover at a time scale reflective of the DOWN state duration (~200 ms). Subthreshold adaptation is a feasible candidate, given that most neurons are depolarized but fire at a very low rate during any given UP state. Adaptation in our model could also include effects of hyperpolarization-activated excitatory processes, such as the h-current. The modeling framework presented here can be used to predict the effects of experimental manipulations of adaptive mechanisms and guide experiments on the biophysical substrates of the neocortical slow oscillation.

Neocortical slow oscillations and hippocampal SWRs are present simultaneously during NREM sleep. Although they appear fundamentally different, our analysis reveals that both can be accounted for using different parameter values of the same model. In the hippocampus, the inter-SWR period is not entirely inactive, but maintains a low rate of spiking. SWR-initiation could come from fluctuations in this low-rate activity or from drive from the entorhinal cortex. The duration of the hippocampal SWR (~60 ms) indicates that the hippocampal adaptive process should activate on a time scale faster than that responsible for recovery from the neocortical DOWN state. Previous work has revealed threshold behavior in the generation of SWRs, indicative

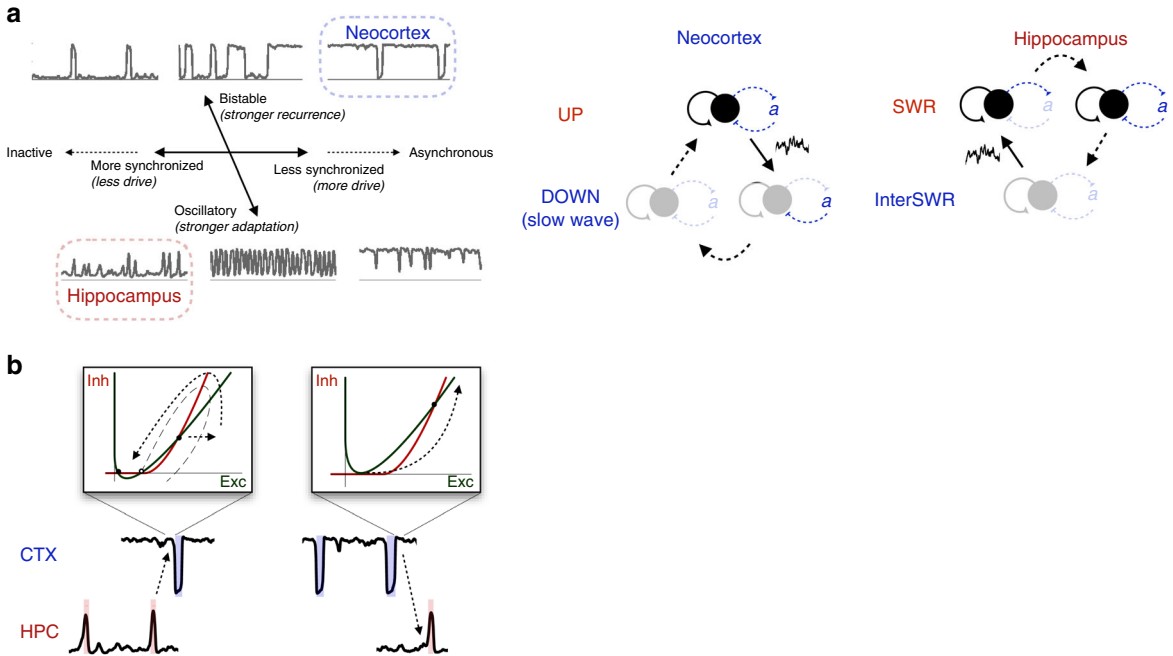

**Fig. 7** Excitable dynamics of NREM sleep. **a** UP/DOWN alternation dynamics are characterized by a spectrum of two effective axes: a degree of synchronization axis determined by the level of drive in the system, and an oscillatory-bistable axis determined by the relative strength of recurrence and adaptation. The tilt of the oscillatory-bistable axis reflects the decrease in the amount of drive needed for equi-duration UP/DOWN states ($I_{1/2}$) as recurrence is increased or adaptation is decreased. (Right) Schematic of the neocortical and hippocampal excitable regimes that produce slow waves and SWRs. Solid arrows indicate noise-induced transitions. **b** Hypothesis for mechanism of SWR/slow wave coupling by excitable dynamics

of Excitable$_{DOWN}$ dynamics, with a GABA$_B$-mediated adaptation mechanism[45,46].

The different nature of recurrent connectivity in the two regions may be responsible for their differing dynamics. Strongly recurrent pyramidal cell populations are found in neocortical layer 5 and the hippocampal CA2 and CA3a subregions[47], the loci of UP state and sharp wave initiation, respectively[48,49]. However, crucial differences exist between connectivity of neocortical layer 5 and hippocampal CA2–3 regions. The neocortex is a modularly organized structure; in contrast, the hippocampus can be conceived as a single expanded cortical module[50]. Excitatory connectivity in layer 5 is local (200 μm), dense (up to 80% connection probability), and follows a 'Mexican hat' excitatory-inhibitory spatial structure with strong local excitatory connections and spatially extensive inhibition[51]. In contrast, excitatory connectivity in the hippocampus is sparse and spatially extensive[47], with local inhibitory connections[52,53]. While layer 5 excitatory synapses are relatively strong, the transmitter release probability of synapses between hippocampal pyramidal neurons is very low, resulting in comparatively weak synapses[54]. Together, these factors indicate that the effective strength of recurrence in the hippocampus is lower than that in neocortex, which would result in the observed DOWN-dominated as opposed to UP-dominated dynamics. To further understand the physiological factors responsible for the distinct NREM dynamics in the two regions will require experimental manipulations that independently manipulate adaptation, recurrent excitation, and excitability.

According to the two-stage model of memory consolidation[55,56], the hippocampus acts as a fast, but unstable, learning system. In contrast, the neocortex acts as a slow learning system that forms long-lasting memories after many presentations of a stimulus. The two-stage model proposes that recently-learned patterns of activity are reactivated in the hippocampus during SWRs, which act as a training signal for the neocortex, and

that the neocortical consolidation of those patterns relies on SWR-slow wave coupling[8,57]. Excitable dynamics provide a mechanism for coordination of slow waves and SWRs (Fig. 7b): the excitatory kick of a hippocampal SWR can induce a neocortical UP→DOWN transition by briefly disrupting the neocortical excitatory/inhibitory balance, while the population burst at the neocortical DOWN→UP transition can induce a hippocampal SWR.

Extensive experimental evidence points towards temporal coordination between slow waves and SWRs. Slow waves in higher-order neocortical regions are more likely following SWRs[8,9], and SWR→slow wave coupling is associated with reactivation in the neocortex[8,57,58]. As is observed in vivo, the ability of transient input to evoke a slow wave in our model is probabilistic, and depends on the input magnitude, local noise, and stability of the UP state. The efficacy of SWR→slow wave induction likely varies by brain state, cortical region, and even SWR spiking content. Further work to investigate how these factors shape SWR→slow wave coupling will shed light on the brain-wide mechanisms of memory consolidation.

How might a SWR-induced slow wave induce changes in the neocortex? Recent work has found that SWR→slow wave coupling alters spiking at the subsequent neocortical DOWN→UP transition[57], which acts a window of opportunity for synaptic plasticity[6,59–61]. In out model, the interaction between excitation and inhibition produces a transient high-frequency oscillation at the DOWN→UP transition. This brief oscillation is reminiscent of the gamma (~60–150 Hz) activity following slow waves in vivo[19] and may act to coordinate and promote plasticity between cell assemblies[62].

In turn, the burst of neocortical activity during the DOWN→UP transition could induce a SWR in the hippocampus. The functional role of slow wave→SWR coupling is less well understood, but hippocampal SWRs are more likely immediately following slow waves in some neocortical regions—

including the entorhinal cortex[8,10]. Slow wave→SWR coupling could provide a mechanism by which neocortical activity is able to bias SWR content, or another mechanism by which the SWR could bias neocortical activity at the DOWN→UP transition. Further, a SWR-slow wave-SWR loop could produce the occasional SWR bursts not captured by our model of hippocampal SWR activity.

The nature of slow wave-SWR interaction reported in studies of dual hippocampal-cortical recordings[8–11] have had differing results: some show that neocortical slow oscillations entrain hippocampal ripples[10,13], while others suggest ripples coincide with neocortical UP→DOWN[8,57] or DOWN→UP[9,63] transitions. Part of this 'controversy' is likely due to the precise location of the recording sites within each of the two structures, and the multiple anatomical paths by which interaction can occur – be it monosynaptic connections, disynaptic connections via the thalamus, connections via the subiculum or via the entorhinal cortex[64]. Our model supports mechanisms for bidirectional interactions between hippocampus and neocortical regions. Elucidating the topological nature of hippocampal-cortical interactions during NREM sleep will require simultaneous recording of neural activity in hippocampus and multiple cortical regions with high spatiotemporal precision. Such future work on regional or state-dependent differences in the directionality of slow wave-SWR coupling will provide insight into the physiological mechanisms that support memory consolidation.

Together, our results reveal that NREM sleep is characterized by structure-specific excitable dynamics in the mammalian forebrain. We found that a model of an adapting recurrent neural population is sufficient to capture a variety of UP/DOWN alternation dynamics comparable to those observed in vivo. The neocortical slow oscillation is well-matched by the model in an Excitable$_{UP}$ regime in which a stable UP state is punctuated by transient DOWN states, while the hippocampal sharp waves are well-matched by the model in an Excitable$_{DOWN}$ regime in which a stable DOWN state is punctuated by transient UP states (Fig. 7a). These complementary regimes of excitable dynamics allow each region to produce characteristic slow wave/SWR events spontaneously or in response to external perturbation. Our results offer a unifying picture of hippocampal and neocortical dynamics during NREM sleep, and suggest a mechanism for hippocampal-neocortical communication during NREM sleep.

## Methods

**Datasets.** The datasets used were reported in Watson et al.[19] (neocortex) and Grosmark and Buzsaki[65] (hippocampus).

For the cortical dataset, silicon probes were implanted in frontal cortical areas of 11 male Long Evans rats. Recording sites included medial prefrontal cortex, anterior cingulate cortex, premotor cortex/M2, and orbitofrontal cortex. Neural activity during natural sleep-wake behavior was recorded using high-density silicon probes during light hours in the animals' home cage. 25 recordings of mean duration 4.8 h ( ± 2.2 std) were recorded. The raw 20 kHz data were low-pass filtered and resampled at 1250 Hz to extract local field potential information. To extract spike times, the raw data high-pass filtering at 800 Hz, and then threshold-crossings were detected. KlustaKwik software was used to cluster spike waveforms occurring simultaneously on nearby recording sites, and Klusters software was used for manual inspection of waveforms consistent with a single neuronal source. Units were classified into putative excitatory (pE) and putative inhibitory (pI) based on the spike waveform metrics. Each animal had 35 ± 12 detected pE units and 5 ± 3 detected pI units (mean ± std over recordings).

For the hippocampal dataset, silicon probes were implanted in the dorsal hippocampus of 4 male Long Evans rats (7 recordings total). Neural activity during sleep was recorded before and after behavior on a linear track. LFP and spikes were extracted similar to the cortical dataset.

**NREM detection.** Sleep state was detected using an automated scoring algorithm as described in Watson et al.[19], with some modifications. As only the NREM state was used in this study, we describe here the process for NREM detection. However, the code for full state detection can be found in the buzcode package (see 'Code availability'). NREM sleep was detected using the FFT spectrogram of a neocortical

LFP channel, calculated in overlapping 10 s windows at 1 s intervals. Power in each time window was calculated for frequencies that were logarithmically spaced from 1 to 100 Hz. The spectral power was then log transformed, and z-scored over time for each frequency. The slow wave power (signature of NREM sleep) was calculated by weighting each frequency by a weight determined from the mean of the weights for the first principal components from the dataset in Watson et al.[19], which was found to distinguish NREM and non-NREM in all recordings. While the same dataset was used here, using the filter (i.e., weighted frequency)-based approach as opposed to PCA makes the algorithm robust for a wider range of recording conditions, especially those in which there is less time spent asleep (and thus NREM may not be expected to account for the largest portion of variance). Like the first principal component, the slow wave filtered signal was found to be bimodal in all recordings, and the lowest point between modes of the distribution was used to divide NREM and non-NREM epochs.

In the hippocampal dataset, manual NREM scoring from Grosmark and Buzsaki[65] was used for this study.

**Slow wave detection.** Slow waves were detected using the coincidence of a two-stage threshold crossing in two signals (Supplementary Fig. 1A, B): a drop in high gamma power (100–400 Hz, representative of spiking[66]) and a peak in the delta-band filtered signal (0.5–8 Hz). The gamma power signal was smoothed using a sliding 80 ms window, and locally normalized using a modified (non-parametric) Z-score in the surrounding 20 s window, to account for non-stationaries in the data (for example due to changes in brain state and noise), that could result in local fluctuations in gamma power. The channel used for detection was determined as the channel for which delta was most negatively correlated with spiking activity, while gamma was most positively correlated with spiking activity.

Two thresholds were used for event detection in each LFP-derived signal, a "peak threshold" and a "window threshold". Time epochs in which the delta-filtered signal crossed the peak threshold were taken as putative slow wave events, with start and end times at the nearest crossing of the window threshold. Peak/window thresholds were determined for each recording individually to best give separation between spiking (UP states) and non-spiking (DOWN states) (Supplementary Fig. 1C). To determine the delta thresholds, all peaks in the delta-filtered signal greater than 0.25 standard deviations were detected as candidate delta peaks and binned by peak magnitude. The peri-event time histogram (PETH) for spikes from all cells was calculated around delta peaks in each magnitude bin, and normalized by the mean rate in all bins. The smallest magnitude bin at which spiking (i.e., the PETH at time = 0) was lower than a set rate threshold (the "sensitivity" parameter, Supplementary Fig. 1D) was taken to be the peak threshold. For example, a sensitivity of 0.5 means that the delta peak threshold is set to the smallest threshold for which spiking drops below 50% of mean spiking activity. The window threshold was set to the average delta value at which the rate crosses this threshold in all peak magnitude bins. The gamma thresholds were calculated similarly, but using drops below a gamma power magnitude instead of peaks above a delta magnitude.

Once the thresholds were calculated, candidate events were then detected in the delta and gamma power signals, and further limited to a minimum duration of 40 ms. Slow wave events were then taken to be overlapping intervals of both the gamma and delta events. DOWN states with spiking above the sensitivity threshold were thrown out.

Detection quality was checked using a random sampling and visual inspection protocol. LFP and spike rasters for random 10 s windows of NREM sleep were presented to a manual scorer, who marked correct SW detections, false alarms, and missed SWs. This protocol was used to estimate the detection quality (miss %, FA %) for each recording (Supplementary Fig. 1E), and to optimize the detection algorithm. 1085–21,147 slow waves (i.e., UP/DOWN states) were detected per recording and used for subsequent analysis. Algorithm for slow wave detection can be found in the buzcode software package.

**SWR detection.** Sharp wave-ripple events were detected using a coincidence of a sharp wave event and a ripple event, with 3134–11,898 SWRs detected per recording and used for subsequent analysis. Sharp wave events were detected as a crossing of a 2.5 std threshold in the 2–50 Hz-filtered signal from the deep hippocampal LFP (below the pyramidal layer). Sharp wave events of duration <20 ms or >500 ms were discarded. Ripple events were detected as a crossing of a 2.5 std threshold in the power of the 80–250 Hz-filtered signal from superficial hippocampal LFP (in the pyramidal layer). Ripple events <25 ms were discarded. Simultaneous ripple/sharp wave events were then merged. Algorithm for SWR detection can be found in the buzcode software package.

**Model implementation.** Phase plane and bifurcation analysis of the model in the absence of noise was implemented in XPP, and a similar code was implemented in MATLAB for simulations of the model with noisy input, for the analysis of UP/DOWN state durations. Noise was implemented using Ornstein-Uhlenbeck noise.

$$d\xi = -\theta\xi dt + \sigma\sqrt{2\theta dt}W_t$$

where $W_t$ is a Weiner process. Time scale $\theta = 0.05$ and standard deviation $\sigma = 0.25$ were used unless otherwise specified.

Simulations of Equations (1–2) and Equations (3–5) were performed in Matlab using the ode45 solver, with input noise $\xi(t)$ pre-computed independently for each simulation using forward Euler method with time step $dt = 0.1$. Accuracy was assessed by comparing results for time steps $dt = 0.1$ and $dt = 0.05$ for a subset of simulations. Statistics for simulations with noise were determined by simulations of duration 60,000 (AU).

A simulated time course was determined to have UP/DOWN states if the distribution of $r(t)$ was bimodal, as determined using a Hartigans dip test[67]. UP/DOWN state transitions were detected as threshold-crossings between high and low rate states. To avoid spurious transition detection due to noise, a "sticky" threshold was used: the threshold for DOWN→UP transitions was taken to be the midpoint between positive crossings of a threshold between the high-rate peak of the rate distribution and the inter-peak trough, while the threshold for UP→DOWN transitions was the midpoint between the low rate peak of the rate distribution and the inter-peak trough.

**UP/DOWN state duration matching**. In vivo and simulated UP/DOWN state durations were compared using a non-parametric distribution matching procedure (Supplementary Fig. 6). Similarity was calculated as

$$s = (1 - KS_{UP}) * (1 - KS_{DOWN})$$

where

$$KS_{UP/DOWN} = \sup_x |F_v(x) - F_s(x)|$$

is the Kolmogorov–Smirnov (KS) statistic, in which $\sup_x$ is the supremum function and $F_{s/v}(x)$ are the empirical cumulative distributions of simulated and in vivo durations. In short, KS measures the largest difference between the observed cumulative distributions for simulated and in vivo durations, where $KS_{UP} = 0$ indicates that the in vivo/simulated UP state durations distributions are identical and $KS_{UP} = 1$ indicates that the in vivo/simulated DOWN state durations distributions are non-overlapping. Similarity is thus bounded between 0 and 1, where $s = 1$ indicates that both UP and DOWN state distributions are identical between simulation and the experimental observation, and $s = 0$ indicates that either the observed UP or DOWN state distributions are non-overlapping with the modeled durations.

There is one free parameter in the fitting procedure, which is $\tau$, the population time constant, or equivalently, the time scale factor from non-dimensionalized model time and seconds. For each simulation, we tested time scale factors from 1 ms to 25 ms with increments of 0.1 ms and used the time scale parameter that gave the highest value for $s$, thus preserving the shapes of the distributions and the relative values of UP/DOWN state durations.

**Reporting summary**. Further information on research design is available in the Nature Research Reporting Summary linked to this article.

## Data availability

All data used in this study are available on the CRCNS database in datasets fcx-1 (crcns. org/data-sets/fcx/fcx-1) and hc-11 (crcns.org/data-sets/hc/hc-11).

## Code availability

The code to reproduce all figures in this study is available at https://github.com/dlevenstein/Levensteinetal2019, and makes use of the buzcode software package: https://github.com/buzsakilab/buzcode.

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

## Acknowledgements

The authors would like to thank Rachel Swanson, William Muñoz, Brendon Watson, Andres Grosmark for discussions during the development of the project and extensive feedback on the manuscript, the NIH training grant for computational neuroscience T90DA043219 for funding and the TPCN trainees for their feedback on the manuscript, NIH U19NS104590-01 for support and Brendon Watson and Andres Grosmark for generously making their data available.

## Author contributions

Conceptualization, D.L., G.B., and J.R.; Methodology, D.L. and J.R.; Formal analysis, D.L. and J.R.; Writing—Original draft, D.L. and J.R.; Writing—Review & Editing, D.L., G.B., and J.R.; Supervision, G.B. and J.R.

## Additional information

**Competing interests:** The authors declare no competing interests.

