## [Peer Review File · Nature Communications]

Reviewers' Comments:

Reviewer #1:

Remarks to the Author:

Levenstein and colleagues have proposed a Wilson and Cowan model to describe cortical and hippocampal dynamics during NREM sleep. The interesting feature of the model is that neither hippocampal nor cortical activity is believed to be intrinsically oscillatory or bistable. Instead, both neuronal populations are believed to stay in a stable state during NREM sleep from which suprathreshold fluctuations induce a population transition which is terminated by adaption (negative feedback). The cortical model is constructed from data previously reported in Watson et al. 2016, whereas the hippocampal model is constructed from data previously reported by Grosmark and Buzsaki 2016. During NREM sleep the cortex is viewed as being in an excitatory up-state whereas the hippocampus is viewed being in an excitatory down-state. This analysis offers a new interpretation of up-down states and suggests a common model for hippocampal and cortical dynamics during NREM sleep.

Conceptual Questions

A feature for both models is transition to a down state or an up state due to "noise" fluctuations. It is interesting that this key feature of the models, namely the transitions from the up to the down states (cortex) or from the down states to the up states (hippocampus) is unexplained. What do the authors consider to be their best guess as to the biophysical instantiation of the "noise"? Do similar noise mechanisms operate for the cortex as well as the hippocampus? This seems like a fundamental feature for helping to establish the ends of the up states and the beginning of the down states for the cortical dynamics and vice versa for the hippocampal dynamics.

Do the authors have experimental data in which there was simultaneous recording of the cortical neurophysiology along with hippocampal neurophysiology to show the two phenomena in the same experimental set up? I would suspect that the mPFC and the CA1 region of the hippocampus have been recorded from simultaneously on several occasions. Could the authors show some of these data?

The current analysis does a compelling job of making plausible the cortical stable up-state model and the hippocampal stable down-state model. What important aspects the experimental data are least well described by a bi-stable state model or an explicit oscillatory model for NREM up-down states? For both models the authors, propose there are nicely defined dwell time distributions taken from the experimental data. This suggests that, at least conceptually, another mathematical possibility is a coupled two-state stochastic oscillator model. Such a model could be easily tuned to give dwell time distributions like the ones shown in figures 1D and 1H. The spike rate could then be just a binary variable that has one value in the up state and a different value in the down state. This form of the spike rate model is what is suggested by plots in figures 1C and 1G.

The data to construct this model come exclusively from rodent experiments. In the Discussion the authors make some links to human NREM sleep architecture. Rodent and human sleep architecture differ appreciably. The evidence that a similar model would hold for human sleep is not as convincing, particularly since many of the features of human sleep are more regular. Could study of human sleep give insight into what might be the biological instantiation of the "noise" processes that are responsible for the up to down transition for the cortical model and the down to up transition for the hippocampal model. Could the bistable or the oscillatory model be more plausible for human NREM sleep?

For these reasons I would suggest a change in the title of the article to

Excitable dynamics of NREM sleep in rodents: a unifying model for the neocortex and the hippocampus.

Minor Points

In general, the paper is well written. A few small points

Should "NREM sleep" be defined as non-rapid eye movement sleep, the first time it is used?

Figure 1 legend. A. "Data was ..." should be "Data were ...".

Reviewer #2:

Remarks to the Author:

In this work the authors present a model of an adapting recurrent neural population incorporating stochastic fluctuations of the input. The model displays a rich repertoire of dynamical regimes which in turn are governed by few key parameters: the recurrent excitation (w), the strength of adaptation (b), and the intensity of the external input drive (I). By fitting the statistics of the duration of the high-firing (UP) and almost quiescent (DOWN) states observed in their experiments, neocortex and hippocampus networks in sleeping (NREM) rats display different excitable dynamical features compatible with the existence of a metastable UP and DOWN state, respectively. An interesting prediction from the model about the state-dependent variation of the statistics of UP/DOWN state duration is successfully tested. Starting from the emerging sensitivity of the two networks to external perturbations, an intriguing hypothesis is discussed about the causal interplay between the two brain structures during the onset of neocortical slow waves and hippocampal sharp wave ripples. The adopted methodology is appropriate and clearly presented. The results appear to be novel and in general both convincing and intriguing by wisely mixing theory and experiments. I am convinced this is an interesting work which can have a wide multidisciplinary audience. My remarks/concerns which should be addressed before publication are listed as follows.

Major points:

1. About the level of adaptation $A_\infty(r)$ at a fixed population rate: why is it sigmoidal? Is there any experimental evidence to motivate this choice? Usually a linear relationship is taken into account. If a linear I/O function would be taken into account, would the richness of dynamical regimes shown by the authors still be visible? To be more specific, I am referring to the dynamical regime represented in Fig. 2Cii.
2. About the similarity between in vivo data and simulations. I have a concern about the way parameters like b , τ_a and σ are identified. How are in Fig. 4B - and hence also in Suppl. Fig. 6 - the values $b = 1$, $\tau_a = 25$ and $\sigma = 0.3$ chosen? If I change τ_a and σ the statistics of the UP and DOWN state duration (Suppl Fig 5) can be dramatically varied. Taking into account such degrees of freedom does the optimal fit of in vivo data become widely degenerate? For instance, in the study about the changes observed at different sleep stages (Fig. 5) it seems that a change in the input drive (I) can explain the observed changes. Actually, I guess that also other pathways in this relative high-dimensional parameter space can account for the excitability change, relying for instance on a suited change in w and b . In addition, what would happen if time scale factor τ_r and τ_a are changed at the same time, is this a way to find equivalent fits? For an example of an alternative trajectory in the bifurcation diagram see Fig. 2 of (Weigenan et al., PLoS Comput. Biol. 2014) [not cited].
3. About the "Effects of balanced excitation and inhibition." In computational neuroscience, the keyword E-I balanced asynchronous state has a specific meaning (see van Vreeswijk & Sompolinsky, Science 1996). In this manuscript no proof is provided that the attractor UP state found in the E-I-A model is expressing a balanced excitation-inhibition regime. Such evidence should be provided or alternatively a different keyword should be used.
4. Uniqueness of the scenario discussed in Fig. 7A. The authors suggest that the different dynamical nature of the hippocampal and neocortical networks during NREM sleep of rodents can be explained by the change of two key parameters: w , the recurrent excitation and I , the input drive. This is surely true, but If I am not wrong another possibility is to consider as key parameter in addition to I the adaptation strength b , instead of w . If I am correct, also this possibility should be mentioned, for

instance referring to papers like (Bazhenov et al., J. Neurosci. 2002; Hill & Tononi, J. Neurophysiol. 2005) [both cited].

5. About the predictions inferred from the model. In the Discussion, one of the main conclusions reported is the characterization of the working dynamical regime of the hippocampus and neocortex (Excitable_UP and Excitable_DOWN, respectively), and how it is related to a difference in the key parameters w and I . Although fascinating, I think this conclusion is only one of at least two possible alternatives. More specifically, I am referring to the possibility that neocortical slow waves can emerge from the interplay with other brain structures like the thalamus (see for instance Sheroziya & Timofeev, J. Neurosci. 2014). This could lead to an alternative explanation of what the authors report, which can be explained assuming a time-varying input $I(t)$ provided by these structures. Under this framework, the statistics of the UP and DOWN durations might be produced by the upstream structure, and the neocortex could have only the role of a "nonlinear" relay station. I think also this alternative non-autonomous network condition should be presented as a possible scenario to test.

Minor points:

1. Fig. 1E-G: the acronym used for sharp-wave ripples in these panels and in the caption is SPW-R, while in the main text and other figures (also in Fig. 1H) is SWR: please use only one acronym if they represent the same thing.
2. About the definition of Excitable_DOWN and Excitable_UP regimes. In Fig. 2 the definition of these two regimes is rather clear: it is the condition when 1 fixed point is stable and other 2 fixed-points are unstable. The former is the one with lowest (highest) population rate r^* for the Excitable_DOWN (Excitable_UP). In the bifurcation diagram I - w shown in Fig. 3D and Suppl. Fig. 3D, these two regimes correspond to the two white flanks near the crossed-pair between oscillatory and bistable regions. Afterwards in the text, Excitable_DOWN and Excitable_UP regimes include also the region where only one stable fixed-point at low or at high population rate exists, respectively. It is absolutely clear from the text that this is because of the presence of the input noise, but in this framework one should explicitly highlight since the beginning that the aforementioned flanks have no special role and that all the white regions are excitable. However, this could not be true for practical reasons. Indeed, the excitability region is limited by the boundary where the input noise has to be so large to elicit a detour from the stable state such that the population rate is so noisy that Up and Down states are no more distinguishable. Could the author be more explicit on that?
3. Fig. 3B: labels like w_{PF} , w_X and w_O should be commented/defined also in the figure caption or in the main text. Now, they are described only in the Suppl. Info.
4. Caption Fig. 3 about panel B: it does not seem to be in the Methods, rather it seems to be in the Suppl. Info.
5. Fig. 3B: y-axis thick label 2 seems to be misplaced. As this bifurcation diagram appears to be the same as Suppl. Fig. 4A, $w = 2$ should be replaced by $w = 3$, if I am not wrong.
6. Reference to "(Figure S4)" should be "(Suppl. Fig. 5)".
7. Experimental measure with errors (for instance those about CV at page 6): I guess errors are standard deviations, but I did not find the number of UP and DOWN states taken into account. It could be useful to know the minimum number of states per animal without referring to the original papers where the data have been collected.
8. In Fig. 4 the number of recordings used ($n = 7$, if I am not wrong) is not mentioned. Writing this would help to understand why the red dots are less in Fig. 4F than the ones shown in Fig. 4B.
9. Looking at the CV_{iSWR} in the hippocampus it has a value significantly greater than 1 in all experiments. This is not compatible with model predictions where maximum CV for the DOWN state is 1, as pointed out by the authors in commenting Fig. 4E and 4G. Could the authors further elaborate on this, for instance in the Discussion where possible limitations of the study are presented?
10. Fig. 5B y-label is not readable in my manuscript version.
11. What is the meaning of " med^{-1}_{NREM} " used as unit measure for the Delta Power in Fig. 5?
12. In the caption of Fig. 5E τ_r is not expressed in ms.

13. Neither Fig. 5E nor Fig. 5F are commented in the text, are they really needed?
14. Pag. 8: Reference to some additional details about the E-I-A model is found as "... (Methods, 26)". However, in the Method section I did not find any additional information about it. Instead, I found the expected Method subsection as Supplemental Info.
15. Pag. 9: about the need of an inhibitory population to stabilize attractors at low firing rates ("However, unlike the excitation-only model ...") I would suggest to cite (Amit & Brunel, Cereb. Cortex 1997) [not cited].
16. Pag. 9: I guess the reference to Fig. 6E and 6F in the main text should be to 6D and 6E, respectively.
17. Pag. 10: Discussion. Not fully clear the meaning of the sentence "This relationship explains the inverse correlation between delta power, measuring mainly the large LFP deflections of the DOWN state, and UP state duration". Delta power increases because LFP deflections during UP/DOWN cycles are larger or because UP/DOWN oscillations are more coherent (less stochastic) and frequent in time. I would suggest to rephrase the sentence.
18. Pag. 21: misprint in the definition of the Ornstein-Uhlenbeck process, instead of dW there should be $W(t)$, a memoryless Wiener process (Gaussian white noise) I guess with infinitesimal mean and variance 0 and 1, respectively. This should be written in the text.
19. The correlation time of the input noise, $1/\theta$, set at 20, seems to be of the order of τ_a . Should not be smaller than τ_a ?
20. About the numerical integration of Eqs (1-5). The numerical approach used by the authors to integrate these two systems - the Matlab `ode45()` function mentioned in the Methods - is appropriate to solve ordinary differential equations (ODEs). However, the systems (1-5) are two sets of stochastic differential equations (SDEs) due to the presence of a noisy input (an OU process ξ is added to the synaptic input). Other methods should be used in this case, for instance the basic Euler-Maruyama scheme or more sophisticated alternatives (see for instance the open source solver "SDETools" available on GitHub as Matlab toolbox). I am rather sure this will not change any of the conclusions reported in the work, however a suited rescaling of the input noise σ may have to be taken into account.
21. Pag. 21. Eqs. [3-5] cited in the Methods are not labeled in the main text. Labels can only be found in the Suppl. Info.
22. Pag. 22: supremum -> supremum.
23. Suppl. Info "General insight...": A reference to an inexistence Fig. 8A is cited.
24. Caption of Suppl. Fig. 5: "Increasing the magnitude of noise increases the duration of stable states", maybe you mean "... noise decreases ...".

Point by point response to referee's comments.

Reviewer #1

- A feature for both models is transition to a down state or an up state due to “noise” fluctuations. It is interesting that this key feature of the models, namely the transitions from the up to the down states (cortex) or from the down states to the up states (hippocampus) is unexplained. What do the authors consider to be their best guess as to the biophysical instantiation of the “noise”? Do similar noise mechanisms operate for the cortex as well as the hippocampus? This seems like a fundamental feature for helping to establish the ends of the up states and the beginning of the down states for the cortical dynamics and vice versa for the hippocampal dynamics.

We agree that noise is a key feature of the models, and that the form and source of noise were underexplored in the original manuscript. We've added the following text to the discussion (lines 286-294, 360-363):

“A key feature of the model is the noise responsible for initiating spontaneous UP->DOWN transitions. In neuronal network modeling, “noise” often refers to unidentified fluctuations in physiological activity. Broadly, biological noise can be divided into fluctuations internal to the population and fluctuations from afferent projections. While we do not explicitly distinguish them in the model, we assume that both sources play a role in initiating cortical UP->DOWN transitions. Population rate fluctuates during the UP state due to finite size effects and temporal correlations that emerge with strong recurrent connections. Similarly, the level afferent activity from thalamo- or cortico-cortical projections would be expected to fluctuate.”

“In the hippocampus, the inter-SWR period is not entirely inactive, but maintains a low rate of activity. SWR-initiating noise could be fluctuations in ongoing low-rate activity during the iSWR period or fluctuations in drive from the entorhinal cortex.”

We also admit, and this is perhaps a more direct answer to the reviewer's question that the term 'noise' often times reflects only our ignorance of or inability to identify/distinguish a dominant mechanism. The mechanisms may be specific to the neocortex and hippocampus or their upstream inputs, and these are not part of our simulations. We agree that this is a very interesting line of further work. However our current general level of mean field modeling will not help distinguish possibilities. The cited rate models used in previous studies of UP/DOWN dynamics also implemented noise in this general additive fashion.

- Do the authors have experimental data in which there was simultaneous recording of the cortical neurophysiology along with hippocampal neurophysiology to show the two phenomena in the same experimental set up? I would suspect that the mPFC and the CA1 region of the hippocampus have been recorded from simultaneously on several occasions. Could the authors show some of these data?

Unfortunately, the datasets available for this study do not have simultaneous frontal cortex/hippocampus recordings. However, the relationship between SWR and SW in similar data has been previously reported and shows increased probability of SWR preceding/following ripples. We've added these citations to the introduction (lines 22-23). We agree that treatment of such data in comparison to the model would be very valuable to understanding their interaction and mechanisms of memory consolidation, and are planning a follow-up study to model cross-areal interaction of the CTX and HPC local networks in comparison with a dataset currently being collected in the Buzsaki lab.

- The current analysis does a compelling job of making plausible the cortical stable up-state model and the hippocampal stable down-state model. What important aspects the experimental data are least well described by a bi-stable state model or an explicit oscillatory model for NREM up-down states? For both models the authors, propose there are nicely defined dwell time distributions taken from the experimental data. This suggests that, at least conceptually, another mathematical possibility is a coupled two-state stochastic oscillator model. Such a model could be easily tuned to give dwell time distributions like the ones shown in figures 1D and 1H. The spike rate could then be just a binary variable that has one value in the up state and a different value in the down state. This form of the spike rate model is what is suggested by plots in figures 1C and 1G.

By coupled two-state stochastic oscillator model we assume that the reviewer means a model in which hippocampus and cortex are both bistable systems that can mutually induce transitions. However, the observed coupling between HPC and cortex is not as strong as would be required for the coupled oscillator model (citations added in line 24). In fact, one of the debated issues in memory consolidation is whether it is the neocortex that entrains sharp wave ripples in the hippocampus or the other way around. Importantly, the bistable/oscillatory models would not give the asymmetry in the CV of UP/DOWN state durations observed in data: bistability would not give short/regular DOWN/SWR durations, and oscillatory would not give irregular UP/iSWR durations.

- The data to construct this model come exclusively from rodent experiments. In the Discussion the authors make some links to human NREM sleep architecture. Rodent and human sleep architecture differ appreciably. The evidence that a similar model would hold for human sleep is not as convincing, particularly since many of the features of human sleep are more regular. Could study of human sleep give insight into what might be the biological instantiation of the “noise” processes that are responsible for the up to down transition for the cortical model and the down to up transition for the hippocampal model. Could the bistable or the oscillatory model be more plausible for human NREM sleep?
For these reasons I would suggest a change in the title of the article to Excitable dynamics of NREM sleep in rodents: a unifying model for the neocortex and the hippocampus.

We respectfully request to keep the title. While it is true that all the data were collected in rodents, we believe that the model should also be relevant for understanding NREM sleep for other species, including humans. The main difference between rodent and human nonREM sleep is the time-dependent change from irregular DOWN states (N2) to relatively rhythmic DOWN states (N3) in humans. compared to the relatively stationary irregularity in the rodent. However, one novel finding from our model is that, we can also demonstrate relatively rhythmic DOWN state in the model and that we do occasionally see rhythmic activity in the rodent. Thus, we suggest that the species difference is mainly quantitative rather than qualitative and that our model will be informative beyond the rodent sleep community. We have removed reference to human sleep structure in the results and added the following text to the discussion (lines 310-314):

“While direct comparison between rodent and human sleep data was not performed, we found a similar evolution in rodent NREM. Quantifying the time spent in these sub-states revealed that the N3-like oscillatory state in the rat occupies only a small fraction of NREM sleep, whereas in humans this stage is more prominent.”

- Minor Points:
 1. Should “NREM sleep” be defined as non-rapid eye movement sleep, the first time it is used?

Yes, we’ve added this (line 17)
 2. Figure 1 legend. A. “Data was ...” should be “Data were ...”.

Fixed

Reviewer #2

- About the level of adaptation $A_{\infty}(r)$ at a fixed population rate: why is it sigmoidal? Is there any experimental evidence to motivate this choice? Usually a linear relationship is taken into account. If a linear I/O function would be taken into account, would the richness of dynamical regimes shown by the authors still be visible? To be more specific, I am referring to the dynamical regime represented in Fig. 2Cii.

With a linear I/O function, the 4 dynamical regimes described would still be available, but no longer in the forms with 2 or 3 unstable fixed points. We’ve added the following text to the supplemental info:
“While we have used a sigmoidal activation function for adaptation, this decision is not crucial for the dynamics described. With a linear activation function we would have been able to get similar results (albeit without the possibility for 5-Fixed point regimes). The choice of a sigmoid activation function was made for two reasons: 1) we found that sigmoid adaptation increases the robustness of the excitable regimes – it decreases the noise required for transitions out of stable fixed points and extends the parameter domain in which excitable alternations are seen. 2) Biologically, adaptation would be expected to saturate; for example, if adaptation were due to a voltage-gated ionic current.”
- About the similarity between in vivo data and simulations. I have a concern about the way parameters like b , τ_a and σ are identified. How are in Fig. 4B - and hence also in Suppl. Fig. 6 - the values $b = 1$, $\tau_a = 25$ and $\sigma = 0.3$ chosen? If I change τ_a and σ the statistics of the UP and DOWN state duration (Suppl Fig 5) can be dramatically varied. Taking into account such degrees of freedom does the optimal fit of in vivo data become widely degenerate? For instance, in the study about the changes observed at different sleep stages (Fig. 5) it seems that a change in the input drive (I) can explain the observed changes. Actually, I guess that also other pathways in this relative high-dimensional parameter space can account for the excitability change, relying for instance on a suited change in w and b . In addition, what would happen if time scale factor τ_r and τ_a are changed at the same time, is this a way to find equivalent fits? For an example of an alternative trajectory in the bifurcation diagram see Fig. 2 of (Weigenan et al., PLoS Comput. Biol. 2014) [not cited].

Indeed, the domain of good fit to in vivo data is highly degenerate (a large domain in parameter space with similar duration statistics and thus high similarity to data). But importantly, it lies primarily within the Excitable regimes. As the reviewer points out, changing the values of other parameters (for example, τ_a and σ) dramatically varies the durations of UP/DOWN durations. However, these changes can generally be compensated by changes in other parameters (for example, the increase in UP state duration by decreasing σ can be compensated by increasing I), or in the time scaling factor. Our approach in light of this degeneracy was to rationalize a number of parameter selections (for example, picking τ_a to reflect the time scale of the neocortical DOWN state; σ was chosen to get sufficient transitions, but not large enough so noise obscures detection of UP/DOWN states), and fit the remaining parameters; with the understanding that further experimental manipulations are necessary (and will hopefully be informed by our study) to narrow down the parameters to a biologically relevant range. We agree this was not made entirely clear in the manuscript and have added Supplemental Figure 7E with durations in I-W space for a few different fixed parameter values of σ , τ , and b , which shows that the exact values of parameters are not important for the conclusion, but the domain of good fit, which is extensive but is only within regimes of excitable dynamics.

As a result of the parameter space degeneracy, the actual parameter values that give the observed sleep stage changes are unclear, for example changes in drive, w , and b , could all give the same effect. We've added the following text to the discussion (lines 315-316):

“Our model predicts that the stages of sleep reflect different stability of the UP state, which may be due to 1) decreased recurrent strength, 2) decreased neuronal excitability or 3) increased strength of adaptation.”

And have included a citation to Weigenan et al, which we appreciate the reviewer bringing to our attention.

- About the “Effects of balanced excitation and inhibition.” In computational neuroscience, the keyword E-I balanced asynchronous state has a specific meaning (see van Vreeswijk & Sompolinsky, Science 1996). In this manuscript no proof is provided that the attractor UP state found in the E-I-A model is expressing a balanced excitation-inhibition regime. Such evidence should be provided or alternatively a different keyword should be used.

We've changed terminology throughout the text to “inhibition-stabilized regime”, and also added citations to van Vreeswijk/Somplinsky as well as Amit/Brunel and emphasized that our model is following Ahmadian et al. 2013.

- Uniqueness of the scenario discussed in Fig. 7A. The authors suggest that the different dynamical nature of the hippocampal and neocortical networks during NREM sleep of rodents can be explained by the change of two key parameters: w , the recurrent excitation and I , the input drive. This is surely true, but if I am not wrong another possibility is to consider as key parameter in addition to I the adaptation strength b , instead of w . If I am correct, also this possibility should be mentioned, for instance referring to papers like (Bazhenov et al., J. Neurosci. 2002; Hill & Tononi, J. Neurophysiol. 2005) [both cited].

We thank the reviewer for suggesting this possibility. We've added the following text to the discussion (lines 368-373, 386-388):

“Our model suggests that a stronger adaptive process in the hippocampus would favor

Excitable_{DOWN} dynamics.

As the relevant parameter is the relative strength of adaptation and recurrence, the different nature of recurrent connectivity in the two regions may also be responsible for their differing dynamics.

To further understand and quantitatively model the physiological factors responsible for the distinct NREM dynamics in the two regions will require experimental manipulations that independently manipulate adaptation, recurrent excitation, and excitability.” .

- About the predictions inferred from the model. In the Discussion, one of the main conclusions reported is the characterization of the working dynamical regime of the hippocampus and neocortex (Excitable_UP and Excitable_DOWN, respectively), and how it is related to a difference in the key parameters w and I . Although fascinating, I think this conclusion is only one of at least two possible alternatives. More specifically, I am referring to the possibility that neocortical slow waves can emerge from the interplay with other brain structures like the thalamus (see for instance Sheroziya & Timofeev, J. Neurosci. 2014). This could lead to an alternative explanation of what the authors report, which can be explained assuming a time-varying input $I(t)$ provided by these structures. Under this framework, the statistics of the UP and DOWN durations might be produced by the upstream structure, and the neocortex could have only the role of a “nonlinear” relay station. I think also this alternative non-autonomous network condition should be presented as a possible scenario to test.

Again, we are grateful for this suggestion. We’ve added the following text to the discussion (lines 294-301):

“We also note that while the isolated cortex can produce UP/DOWN state alternations³⁶, we should consider the thalamocortical system for an understanding of slow wave dynamics in vivo³⁷. Because the cortex and corresponding thalamic nuclei are highly interconnected, cortex and thalamus may transition UP and DOWN together and reflect interacting (as opposed to independent) systems. However, it was recently found that cortex tends to lead the thalamus into the DOWN state³⁸. Future work should expand the model to include a thalamic population, which would also allow a better understanding of the interaction of slow waves with thalamocortical spindle oscillations^{8,39,40”}

When in the future we expand the model and include the thalamus, this will be an excellent issue to address.

- Minor points
 1. Fig. 1E-G: the acronym used for sharp-wave ripples in these panels and in the caption is SPW-R, while in the main text and other figures (also in Fig. 1H) is SWR: please use only one acronym if they represent the same thing.

We thank the reviewer for catching this inconsistency, SWR is now used throughout.

2. About the definition of Excitable_DOWN and Excitable_UP regimes. In Fig. 2 the definition of these two regimes is rather clear: it is the condition when 1 fixed point is stable and other 2 fixed-points are unstable. The former is the one with lowest (highest) population rate r^* for the Excitable_DOWN (Excitable_UP). In the bifurcation diagram I-w shown in Fig. 3D and Suppl. Fig. 3D, these two regimes correspond to the two white flanks near the crossed-pair between oscillatory and bistable regions. Afterwards in the text, Excitable_DOWN and Excitable_UP regimes include also the region where only one

stable fixed-point at low or at high population rate exists, respectively. It is absolutely clear from the text that this is because of the presence of the input noise, but in this framework one should explicitly highlight since the beginning that the aforementioned flanks have no special role and that all the white regions are excitable. However, this could not be true for practical reasons. Indeed, the excitability region is limited by the boundary where the input noise has to be so large to elicit a detour from the stable state such that the population rate is so noisy that Up and Down states are no more distinguishable. Could the author be more explicit on that?

We agree this was unclear in the original text. We have changed the text for Figure 2C: “Four UP/DOWN regimes available to the model, as distinguished by location of stable fixed points (see also Supplemental Figure 3)” to match the main text that the location of the single stable fixed point, is the defining factor.

3. Fig. 3B: labels like w_{PF} , w_X and w_O should be commented/defined also in the figure caption or in the main text. Now, they are described only in the Suppl. Info.

Definitions for these labels have been added to Fig 3B caption.

4. Caption Fig. 3 about panel B: it does not seem to be in the Methods, rather it seems to be in the Suppl. Info.

Fig 3b caption has been updated to refer to Supplemental Info

5. Fig. 3B: y-axis thick label 2 seems to be misplaced. As this bifurcation diagram appears to be the same as Suppl. Fig. 4A, $w = 2$ should be replaced by $w = 3$, if I am not wrong.

We thank the reviewer for catching this. Y-axis label has been corrected.

6. Reference to “(Figure S4)” should be “(Suppl. Fig. 5)”.

This has been corrected. (Line 132)

7. Experimental measure with errors (for instance those about CV at page 6): I guess errors are standard deviations, but I did not find the number of UP and DOWN states taken into account. It could be useful to know the minimum number of states per animal without referring to the original papers where the data have been collected.

The errors are standard deviations of CV over recordings. We've added the following text to the methods (Lines 626, 685-686):
“3134-11898 SWRs detected per recording and used for subsequent analysis”
“1,085 - 21,147 slow waves (i.e. UP/DOWN states) were detected per recording and used for subsequent analysis.”

8. In Fig. 4 the number of recordings used ($n = 7$, if I am not wrong) is not mentioned.

Writing this would help to understand why the red dots are less in Fig. 4F than the ones shown in Fig. 4B.

We added the following text to the methods (Line 623): “7 recordings total”

9. Looking at the CV_iSWR in the hippocampus it has a value significantly greater than 1 in all experiments. This is not compatible with model predictions where maximum CV for the DOWN state is 1, as pointed out by the authors in commenting Fig. 4E and 4G. Could the authors further elaborate on this, for instance in the Discussion where possible limitations of the study are presented?

We've added the following text to the discussion (Lines 423-425):
“a SWR-slow wave-SWR loop could produce the occasional SWR bursts not captured by our model of hippocampal SWR activity in isolation”, and explicitly pointed out CV >1 in line 194

10. Fig. 5B y-label is not readable in my manuscript version.

Fixed

11. What is the meaning of “med⁻¹_NREM” used as unit measure for the Delta Power in Fig. 5?

We've added the following text to the figure legend “delta power normalized to median power during NREM sleep”

12. In the caption of Fig. 5E τ_r is not expressed in ms.

τ_r in the caption for the panel (Now Supplemental 8F) now indicates $\tau_r=5ms$.

13. Neither Fig. 5E nor Fig. 5F are commented in the text, are they really needed?

We agree, 5E has been moved to supplemental figure 8, 5F has been removed.

14. Pag. 8: Reference to some additional details about the E-I-A model is found as “... (Methods, 26)”. However, in the Method section I did not find any additional information about it. Instead, I found the expected Method subsection as Supplemental Info.

We fixed the reference (line 238)

15. Pag. 9: about the need of an inhibitory population to stabilize attractors at low firing rates (“However, unlike the excitation-only model ...”) I would suggest to cite (Amit & Brunel, Cereb. Cortex 1997) [not cited].

We've added citations to Amit/Brunel as well as van Vreeswijk/Somplinsky and Ahmadian et al. 2013.

16. Pag. 9: I guess the reference to Fig. 6E and 6F in the main text should be to 6D and 6E, respectively.

We've fixed the references (Line 248)

17. Pag. 10: Discussion. Not fully clear the meaning of the sentence "This relationship explains the inverse correlation between delta power, measuring mainly the large LFP deflections of the DOWN state, and UP state duration". Delta power increases because LFP deflections during UP/DOWN cycles are larger or because UP/DOWN oscillations are more coherent (less stochastic) and frequent in time. I would suggest to rephrase the sentence.

We've removed this sentence when re-writing the section to address other comments.

18. Pag. 21: misprint in the definition of the Ornstein-Uhlenbeck process, instead of dW there should be $W(t)$, a memoryless Wiener process (Gaussian white noise) I guess with infinitesimal mean and variance 0 and 1, respectively. This should be written in the text.

We've specified W_t as a weiner process (Line 694) and used the notation following Ermentrout and Termen.

19. The correlation time of the input noise, $1/\theta$, set at 20, seems to be of the order of τ_a . Should not be smaller than τ_a ?

We've added Supplemental Figure 7E, which shows that the choice of τ_a (and thus its relationship to θ) is not critical for our results.

20. About the numerical integration of Eqs (1-5). The numerical approach used by the authors to integrate these two systems - the Matlab `ode45()` function mentioned in the Methods – is appropriate to solve ordinary differential equations (ODEs). However, the systems (1-5) are two sets of stochastic differential equations (SDEs) due to the presence of a noisy input (an OU process ξ is added to the synaptic input). Other methods should be used in this case, for instance the basic Euler-Maruyama scheme or more sophisticated alternatives (see for instance the open source solver "SDETools" available on GitHub as Matlab toolbox). I am rather sure this will not change any of the conclusions reported in the work, however a suited rescaling of the input noise σ may have to be taken into account.

We used the Euler method to pre-compute the noise (ξ), which was then used as a time-varying input to Eqns 1-5 with `ode45()`. We've updated the methods section (line 697) for clarity.

21. Pag. 21. Eqs. [3-5] cited in the Methods are not labeled in the main text. Labels can only be found in the Suppl. Info.

We've added the labels to the main text (lines 236-238).

22. Pag. 22: supremium -> supremum.

Fixed

23. Suppl. Info “General insight...”: A reference to an inexistence Fig. 8A is cited.

Removed

24. Caption of Suppl. Fig. 5: “Increasing the magnitude of noise increases the duration of stable states”, maybe you mean “... noise decreases ...”.

Fixed

We thank the reviewers for their attention to detail.

Reviewers' Comments:

Reviewer #1:

Remarks to the Author:

The authors have nicely responded to my critique.

1. The model would be more credible if the authors could quantify the magnitudes of the noise sources. The noises and their magnitudes are critical for the model. Are their data or modeling studies that suggest the magnitude of these noise sources? It is unsatisfying to have this crucial element of the framework left not well constrained.
2. The authors state that they plan to do simultaneous recordings from PFC and hippocampus. Perhaps I am mistaken. However, I would have thought that investigators in the hippocampal field had conducted such recordings. Showing the stated phenomena in actual data would make the model far more compelling. At the moment, the current model is quite elegant but primarily a theoretical framework that makes a strong, compelling prediction.
3. As regards the article title, human sleep and rodent sleep are not just quantitatively but qualitatively different. REM-NREM cycling in humans as well as sleep-wake cycling in humans differ quite a bit from their counterparts in rodents. A model which accurately describes rodent sleep remains extremely valuable. The title of the paper should reflect the scope of the work. That the current model extends directly to humans is not clear given the qualitative between species differences.

Reviewer #2:

Remarks to the Author:

The authors have fully addressed all my concerns and added a new figure as supplemental info. This figure supports a widely revised Results subsection "Inhibition stabilized a low-rate UP state..." where the authors improved the description of the possible mechanism underlying the cortico-hippocampal interaction giving rise to the interplay between SWR and slow waves.

This improved version of the manuscript is in my opinion well suited to be published in Nat Comm. I would only suggest few minor changes in part related to my previous remarks, which do not require any further check from my side:

1. About the sigmoidal shape $A_{\infty}(r)$, I fully agree with authors that "biologically, adaptation would be expected to saturate", however, I would suggest to include part of the motivations added to the supplemental info into the main text.
2. At line 292 of the Discussion, after "due to finite size effects" I would suggest to add a citation to (Mattia & Sanchez-Vives, 2012 – cited), as similarly to (Schwalger et al., 2017) also in that paper the population discharge rate was described as a fluctuating variable due to the finite number of neurons in the network.
3. At line 316 of the Discussion, after "increased strength of adaptation" I would suggest to cite (Destexhe, 2009 – cited) together with (Weigenand et al., 2017), as that is another paper where a change of adaptation strength was suggested to drive state transitions in spiking neuron networks.
4. About my minor point 18, surely I have not been able to explain myself properly. My remark was related to the fact that "dW_t" seems to be a differential in time of a Wiener process. Actually, W_t should be directly used (and I guess this is exactly what the authors have done), such that at each time step of the numerical integration a random number is generated from a Gaussian distribution

with zero mean and unit variance, and eventually multiplied by $\sigma \sqrt{2 \theta dt}$ and added to $-\theta \psi dt$ to obtain $d\psi$ (see Cox & Miller, The theory of stochastic processes, 1965).

5. At page 10 of supplemental info "One the other hand" should be "On the other hand."

6. In the supplemental info "Ahmadian and Milller (2013)" is mentioned but the related paper is not listed among the References.

7. In the first line of the Supplemental Figure 4 caption "drive drive" should be "drive."

Reviewer #1 (Remarks to the Author):

The authors have nicely responded to my critique.

We thank the reviewer for his/her acknowledgement.

1. The model would be more credible if the authors could quantify the magnitudes of the noise sources. The noises and their magnitudes are critical for the model. Are their data or modeling studies that suggest the magnitude of these noises sources? It is unsatisfying to have this crucial element of the framework left not well constrained.

We agree that the biological sources and features of “noise” are interesting aspects of this study, but we do not think that our current treatment lessens its credibility, or that a more extensive treatment is within its scope.

Specifically, we unable to quantify the magnitude of the noise sources. Noise in our model reflects a combination of fluctuating drive from various sources – in the discussion (Lines 292-297, 340-343) we have mentioned external sources (i.e. fluctuations in the level of drive from afferent cortico-cortical and thalmo-cortical projects), and internal sources that emerge from ongoing population activity (finite size effects and temporal correlations). Neither at our level of modeling nor in our data do we have a way to distinguish or measure these sources. Quantification and incorporation of distinct sources are ongoing challenges in mean field modeling with noise, and we have treated noise similarly to previous work. The cited rate models used in previous studies of UP/DOWN dynamics also implemented noise in this general additive fashion in which the noise magnitude was chosen (“arbitrarily”) to account for statistics of transitions. We have given a general description of how the dynamics of UP/DOWN alternations are influenced by noise (Supp Fig 5,7), which gives intuition of how alternation properties should change as the noise properties change.

We also note that we have shown that the magnitude of the noise is *not critical for the main finding* (Supp Fig 7) – namely that HPC and CTX are in excitable regimes during NREM sleep. Our results (excitable dynamics) are robust to a range of noise magnitudes (Supp Fig 7), indicating the absolute value of this parameter is not crucial for the findings, except that it must be sufficiently large to evoke transitions and sufficiently small to not overwhelm the difference between UP/DOWN states. We’ve shown that a range of parameter values are acceptable, and how changes in those parameters can be compensated for by changes in other parameters... for example decreasing the noise magnitude could be compensated for by decreasing the depth of the basin of attraction of the stable state – say by decreasing the tonic drive or decreasing the recurrent strength.

2. The authors state that they plan to do simultaneous recordings from PFC and hippocampus. Perhaps I am mistaken. However, I would have thought that investigators in the hippocampal field had conducted such recordings. Showing the stated phenomena in actual data would make the model far more compelling. At the moment, the current model is quite elegant but primarily a theoretical framework that makes a strong, compelling prediction.

The reviewer is correct that simultaneously recordings from PFC (or other cortical areas) and hippocampus have been reported. However, the nature of interaction varies across these reports. Some reports show how slow oscillations entrain hippocampal ripples, whereas others how ripples affect the UP and DOWN states. Part of this 'controversy' is likely due to the precise location of the recording sites within each of the two structures. To do justice to the Reviewer's question will require recordings from the hippocampus and multiple cortical areas and beyond the scope of this study. As suggested by the reviewer, in the revised manuscript we reference experimental findings (see next paragraph), which are relevant to our modeling predictions and illustrate the complex topological nature of hippocampal-cortical interaction.

We've added the following text to the discussion (lines 405-418):

As illustrated above, the nature of slow wave-SWR interaction reported in studies of dual hippocampal-cortical recordings⁸⁻¹¹ have had differing results: some show that neocortical slow oscillations entrain hippocampal ripples^{10,13}, while others suggest ripples coincide with neocortical UP->DOWN^{8,57} or DOWN->UP^{9,63} transitions. Part of this 'controversy' is likely due to the precise location of the recording sites within each of the two structures, and the multiple anatomical paths by which interaction can occur – be it monosynaptic connections, disynaptic connections via the thalamus, connections via the subiculum or via the entorhinal cortex⁶⁴. Our model represents a general framework for the study of excitable dynamics and suggests explicit mechanisms by which bidirectional interactions could occur between hippocampus and neocortical regions. Elucidating the topological nature of hippocampal-cortical interactions during NREM sleep will require novel methods that allow simultaneous recording in hippocampus and multiple cortical regions with high spatiotemporal precision. Such future work on regional or state-dependent differences in the directionality of slow wave-SWR coupling will provide insight into the physiological mechanisms that support memory consolidation.

3. As regards the article title, human sleep and rodent sleep are not just quantitatively but qualitatively different. REM-NREM cycling in humans as well as sleep-wake cycling in humans differ quite a bit from their counterparts in rodents. A model which accurately describes rodent sleep remains extremely valuable. The title of the paper should reflect the scope of the work. That the current model extends directly to humans is not clear given the qualitative between species differences.

We agree with the reviewer that rodent and human sleep differs in many respects. We thank the reviewer for pointing out the need for clarity on the use of rodents as a model system for NREM sleep dynamics. While our matching to data was in rats, the model itself is more general than rat sleep, or even sleep alternations alone. We thus worry that having rodent/rat in the title will lessen readership with human sleep researchers. We've made the use of the rodent as our model system more clear, early in the text (abstract), and have added text to the discussion as per the need for more comparative studies.

(line 7) "experimental observations of naturally sleeping rats"

(line 35-40) "While rodent sleep does differ from that seen in humans, both humans and rodents have NREM sleep of varying "stages" or "depths" that correspond to changes in slow wave dynamics. Due to these similarities and the accessibility of invasive high density electrophysiological data, the rodent has been a model system for studying the internally-organized dynamics of the sleeping brain."

(line 43) "match experimental data from NREM sleep in naturally-sleeping rats"

(line 307) "We found that the depth of NREM sleep in the rodent"

(line 319) "light NREM sleep in the human (stage N1)"

(line 321-323) "Further comparative studies on the differences between rodent and human sleep are needed, and we hope that our model can provide a framework to guide such future work."

Reviewer #2 (Remarks to the Author):

The authors have fully addressed all my concerns and added a new figure as supplemental info. This figure supports a widely revised Results subsection "Inhibition stabilized a low-rate UP state..." where the authors improved the description of the possible mechanism underlying the cortico-hippocampal interaction giving rise to the interplay between SWR and slow waves.

This improved version of the manuscript is in my opinion well suited to be published in Nat Comm. I would only suggest few minor changes in part related to my previous remarks, which do not require any further check from my side:

1. About the sigmoidal shape $A_\infty(r)$, I fully agree with authors that "biologically, adaptation would be expected to saturate", however, I would suggest to include part of the motivations added to the supplemental info into the main text.

In the interest of conserving space, and readability for non-mathematically oriented readers, we've left the discussion of the functional choice of adaptation in the supplemental. However, we've made clearer that the discussion is there:

Line (75-77): ...however we note that this choice is not critical for the generality of our findings. Further discussion of model details and physiological interpretation of model parameters can be found in the Supplemental Info.

2. At line 292 of the Discussion, after "due to finite size effects" I would suggest to add a citation to (Mattia & Sanchez-Vives, 2012 – cited), as similarly to (Schwalger et al., 2017) also in that paper the population discharge rate was described as a fluctuating variable due to the finite number of neurons in the network.

We thank the reviewer for the citation suggestions and have added them.

3. At line 316 of the Discussion, after "increased strength of adaptation" I would suggest to cite (Destexhe, 2009 – cited) together with (Weigenand et al., 2017), as that is another paper where a change of adaptation strength was suggested to drive state transitions in spiking neuron networks.

Added

4. About my minor point 18, surely I have not been able to explain myself properly. My remark was related to the fact that " dW_t " seems to be a differential in time of a Wiener process. Actually, W_t should be directly used (and I guess this is exactly what the authors have done), such that at each time step of the numerical integration a random number is generated from a Gaussian distribution with zero mean and unit variance, and eventually multiplied by $\sqrt{2 \theta dt}$ and added to $-\theta \xi dt$ to obtain $d\xi$ (see Cox & Miller, The theory of stochastic processes, 1965).

Fixed

5. At page 10 of supplemental info "One the other hand" should be "On the other hand."

Thanks, changed

6. In the supplemental info "Ahmadian and Milller (2013)" is mentioned but the related paper is not listed among the References.

Oops, added

7. In the first line of the Supplemental Figure 4 caption "drive drive" should be "drive."

Oops, fixed

Reviewers' Comments:

Reviewer #1:

Remarks to the Author:

Researchers in the rodent sleep community readily acknowledge the difference between human and rodent sleep. It without additional data it is not clear how much insight this paper gives about human sleep. The description of the rodent sleep is nevertheless extremely valuable. I think the article should be titled to reflect that fact that it gives insights into rodent sleep and not human sleep.